# DiSC: Differential Spectral Clustering of Features

**Ram Dyuthi Sristi, Gal Mishne**
UC San Diego
La Jolla, CA, USA
{rsristi, gmishne}@ucsd.edu

**Ariel Jaffe**
Hebrew University
Jerusalem, Israel
ariel.jaffe@mail.huji.ac.il

## Abstract

Selecting subsets of features that differentiate between two conditions is a key task in a broad range of scientific domains. In many applications, the features of interest form clusters with similar effects on the data at hand. To recover such clusters we develop DiSC, a data-driven approach for detecting groups of features that differentiate between conditions. For each condition, we construct a graph whose nodes correspond to the features and whose weights are functions of the similarity between them for that condition. We then apply a spectral approach to compute subsets of nodes whose connectivity pattern differs significantly between the condition-specific feature graphs. On the theoretical front, we analyze our approach with a toy example based on the stochastic block model. We evaluate DiSC on a variety of datasets, including MNIST, hyperspectral imaging, simulated scRNA-seq and task fMRI, and demonstrate that DiSC uncovers features that better differentiate between conditions compared to competing methods.

## 1  Introduction

Detecting variables or features that separate between two or more conditions is a critical task in many scientific domains. Often, the separation between conditions is caused by a large number of strongly dependent features that form one or more differentiating clusters. Recovering those clusters can provide insights into the data and its underlying mechanisms.

Strongly dependent differentiating features are common, for example, in transcriptome analysis, where the data consists of the gene expression of multiple cells. Often, the cells correspond to two or more biological conditions, such as various cell types, the existence of a particular disease, response to medical treatment, or two-time points in an evolutionary process [48].

Uncovering groups of differentiating genes (also known as *pathways*) may significantly contribute to our understanding of biological processes happening in one condition and not in the other. Differential feature grouping has various other interesting applications in neuroscience [43], computer vision and machine learning [6]. In all applications, detecting groups that are significant for differentiating between conditions adds interpretability to the otherwise black box models.

Due to its importance in various domains, the feature selection task has been the focus of intensive research. Often, methods for feature selection are designed to yield a sparse output, for example, by selecting only a small number of features out of a large correlated group. One example is the best subset selection approach, where features are iteratively added or removed until a predetermined criterion is met. Another popular approach is to achieve a sparse output by adding regularization terms to a linear (or general linear) model that predicts a function over the data points, e.g., class label. For example, classical LASSO relaxes the $l_0$ norm of the coefficient vector with its convex surrogate $l_1$ [13, 37]. Alternatively, a recent approach aims to approximate the $l_0$ regularizer via stochastic gates [44].

To address the case of correlated features, various extensions of LASSO were developed. Clustered LASSO [33] promotes sparsity among the coefficients and equal significance to non-zero coefficients

36th Conference on Neural Information Processing Systems (NeurIPS 2022).

in the same group. Ordered Weighted L1 regularization (OWL) [4] penalizes the coefficients in the order of their magnitude, i.e., the higher the absolute value of the coefficient, the higher the penalty that will be imposed. This induces equal coefficients to the correlated features [14]. Elastic Net [49] uses a combination of $l_1$ and $l_2$ regularizers which creates a feature grouping effect. Cluster Elastic Net [42] assumes that each feature belongs to one of k-distinct clusters and uses a clustering penalty, along with LASSO, which minimizes the sum of pairwise distances between the associations of features with the prediction within each cluster. A method that combines feature selection and grouping was also developed in [5]. A few methods explicitly use the correlation between the features in the regularization terms. [25] estimates the covariance between features from which they form a graph. The graph Laplacian matrix is then used as a quadratic regularizer function.

This line of work typically addresses a linear regression model. However, in many settings the dependency on the important features is highly non-linear. This is typically the case, for example, in most discrete settings such as classification and clustering. Furthermore, most of these methods do not separate different groups of correlated features that are significant in a classification or regression problem. In many settings, such a separation is important to gain insight into different sources of the variability in the data.

A different approach related to this work is feature selection via discriminant subspace analysis methods such as linear discriminant analysis [36, 32] and the Fukunaga-Koontz transform [17, 31]. Here, the primary goal is to identify the subspace that best discriminates the classes according to a given criterion. For example, the classic Fischer criterion [16] maximizes the ratio between inter-class variance and intra-class variance. Recent works applied similar methods in the context of non-negative and Boolean matrix factorization [20, 21] . Some metric learning approaches [28], which take into account triplet relationships between points, learn a linear transform on the features so that the distance in the projected space better separates the classes. [7] introduces a Riemannian geometry based composite kernel for differential feature extraction. However, these approaches do not explicitly perform feature selection to identify different groups of correlated features that separate the classes.

In this paper we develop a data-driven method to detect and group relevant features that does not rely on a specific model. We consider the classification setting, where our aim is to identify class-specific information in the feature space that distinguishes between different conditions. To that end, we develop DiSC, a data-driven method to reveal groups of differential features. Our approach consists of two main steps: first, for each class, we compute a class-specific graph, whose nodes correspond to the features of that class separately. Next, we apply a spectral approach to obtain an embedding of the features that identifies groups of features whose connectivity differs between the graphs. Our contributions in this paper are:

1. We develop a data-driven spectral approach on the feature space to identify groups of correlated features that distinguish between datasets

2. Our solution is non-symmetric such that we identify class-specific differential features

3. We provide theoretical analysis in the setting of a stochastic block model underlying the features

Our spectral approach is related to recent papers that address the challenges of multiview and data fusion. Here, data samples are observed by multiple sets of sensors, and the goal is to identify latent representations of the samples that are shared for all sets. The shared information can be recovered via the shared latent space between multiple sources. Alternating Diffusion Maps (ADM) [24] is a nonlinear manifold learning approach to reveal shared latent variables. Shnitzer et al. [35] extend this to identifying both common structures and the differences between the manifolds underlying the different modalities. In an alternative solution, [26] proposes a kernel and distance metric for diffusion maps on multiview datasets. There are numerous other data fusion techniques [27], [10] and applications [22]. Most of these techniques require one-to-one correspondence between data samples from different views, e.g., simultaneous recordings from different sources, and they mainly focus extracting a shared subspace or a shared hidden variable of the samples. In contrast, in our setting the correspondence is between features in two datasets, and not between samples. Our goal is to uncover connectivity patterns that are *condition specific*.

## 2 Problem formulation

We begin with a formal description of our problem, followed by relevant notation. We consider two datasets $X^A \in \mathbb{R}^{n_A \times p}$ and $X^B \in \mathbb{R}^{n_B \times p}$ where the rows in each matrix are high-dimensional samples and the columns are the features, see illustration in Fig. 1a. Both datasets have the same $p$ features, such that the feature column in $A$, denoted $X_{\cdot i}^A \in \mathbb{R}^{n_A}$ corresponds to the feature column in $B$, denoted $X_{\cdot i}^B \in \mathbb{R}^{n_B}$. Our goal in this work is to detect one or more *subsets of features* that together differentiate between the two states $A$ and $B$. These subsets can be, for example, groups of genes that form a biological pathways, or a subset of brain parcels in fMRI data with similar blood oxygen level dependent (BOLD) activity.

In our approach, we compute two graphs, denoted $G_A, G_B$ with $p$ nodes that correspond to the features of the given data (Fig. 1b). We denote by $W_A, W_B$ the weight matrices of the two graphs, whose elements are functions of the similarity between features, as computed by two kernel functions,

$$K_A(X_{\cdot i}^A, X_{\cdot j}^A) : \mathbb{R}^{n_A} \times \mathbb{R}^{n_A} \to \mathbb{R} \qquad \text{and} \qquad K_B(X_{\cdot i}^B, X_{\cdot j}^B) : \mathbb{R}^{n_B} \times \mathbb{R}^{n_B} \to \mathbb{R}. \quad (1)$$

These for example can be the RBF kernel $K(x, x') = \exp\{-\|x - x'\|^2/\epsilon^2\}$. Our underlying assumption is that differences between the two states $A, B$ are expressed as differences in connectivity patterns between the two graphs $G_A$ and $G_B$. For example, a pair of features $i, j$ may be insignificant in state $A$ and highly significant and dependent in $B$. This will imply,

$$K_A(X_{\cdot i}^A, X_{\cdot j}^A) \approx 0 \quad \text{and} \quad K_B(X_{\cdot i}^B, X_{\cdot j}^B) > 0.$$

If there is a subset of $l$ features $i_1, \ldots, i_l$ that are strongly dependent in $B$ but not in $A$, the resultant nodes in $G_B$ will form an independent component with dense connections among the nodes. However, this independent component will not exist in $G_A$, see illustration in Figure 1. Thus, the task of obtaining sets of significant features boils down to identifying the independent components that appear in one graph but not the other.

## 3 DiSC

### 3.1 A graph cut perspective

Given a graph $G$ with $n$ nodes and its associated weight matrix $W \in \mathbb{R}^{n \times n}$, the minimum-cut of $G$ is the minimum, over all possible partitions $\alpha$ and $\beta$, of the sum of the edge weights $\sum_{i \in \alpha, j \in \beta} W(i, j)$. This task is strongly related to the spectral clustering algorithm. For completeness, we begin with a brief description of this relation. For a more thorough review see [41].

A variation of the minimum-cut, designed to avoid highly imbalanced partitions, is the ratio-cut

$$\text{Rcut}(\alpha, \beta) = \sum_{i \in \alpha, j \in \beta} W_{ij} \left( \frac{1}{|\alpha|} + \frac{1}{|\beta|} \right), \quad (2)$$

where $|\alpha|, |\beta|$ denote the subset size. This can be formulated as

$$\text{Rcut}(\alpha, \beta) = f^T L_u f, \qquad \text{where } f_i = \begin{cases} \sqrt{\frac{|\beta|}{|\alpha|}} & i \in \alpha \\ -\sqrt{\frac{|\alpha|}{|\beta|}} & i \in \beta, \end{cases} \quad (3)$$

and $L_u$ is the unnormalized graph Laplacian matrix given by $D - W$, where $D$ is a diagonal the degrees of each node in its diagonal. Note that $f$ is a non-binary indicator vector where each elemnt of f, represented by $f_i$ only takes one of the two different values as given in Eq.(3). If $\alpha, \beta$ are non-empty, then $f$ is orthogonal to the constant vector with $\|f\|_2^2 = n$. Minimizing Eq. (3) over all partitions is a discrete optimization problem. To avoid it, one can relax the discrete requirement over the elements of $f$ in Eq. (3), while still maintaining the constraints $f^T 1 = 0$ and $\|f\|_2^2 = n$. This relaxation yields a simple spectral solution, where the nodes are partitioned according to the sign of the graph Laplacian eigenvector corresponding to the second smallest eigenvalue (a.k.a Fiedler vector).

The minimum-cut criterion is designed to attain densely connected components that have low connectivity with the rest of the graph. In our work, however, densely connected components that appear in both graphs are of no interest, as they do not differentiate between the two states. Rather,

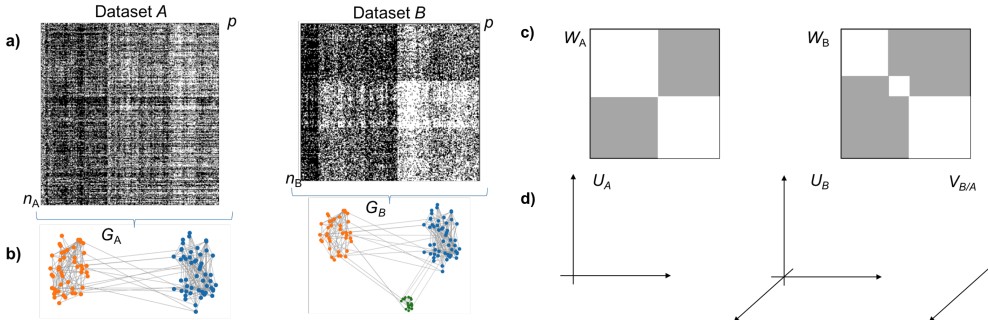

Figure 1: DiSC overview. a) Datasets $A$ and $B$ share the same $p$ features. b) Constructing graphs $G_A$ and $G_B$ on the feature space for each dataset, we aim to find which nodes have different connectivity pattern between the two datasets. c) The weight matrices each follow a stochastic block model, where in $B$ a subset of of the nodes form a separate block. d) Subspaces spanned by the eigenvectors of $P_A$ and $P_B$, and the differential feature between $B$ and $A$.

we would like to detect a partition $(\alpha, \beta)$, say on $G_B$, that has two properties: 1) Rcut$(\alpha, \beta)$, on $G_B$ is minimized, which indicates that $\alpha$ and $\beta$ are independent connected components in $G_B$; 2) To avoid detection of components that exist both in $G_A$ and $G_B$, Rcut$(\alpha, \beta)$ on $G_A$ is lower bounded by some constant. This double objective can be formulated by

$$\min_f f^T L_B f \qquad \text{s.t.} \quad f^T L_A f \geq \gamma, \quad \|f\| = 1. \tag{4}$$

The constraint $\|f\| = 1$ is added for uniqueness, since $L_A, L_B$ are rank deficient. Relaxing the discrete requirement in Eq. (3), the solution to Eq. (4) is given by the generalized eigenvalue problem $L_B f = \mu L_A f$. Since $L_A$ is not invertible, one numerical trick is to strengthen the diagonal of $L_A$ to make it a full rank matrix. Relaxing the discrete requirement in (3), the solution is given by the eigenvalue problem,

$$(L_A + \epsilon I)^{-1} L_B f = \mu f. \tag{5}$$

Motivated by the above derivation, in the next section we present our spectral approach for group feature discovery. In the appendix we present a similar derivation that is based on the normalized cut, whose relaxation relates to the eigenvectors of the random walk graph Laplacian.

### 3.2 Spectral approach for group differential feature extraction

Our proposed approach for detecting differences between graphs is motivated by the solution of the double objective criterion given in Eq. (5). However, due to the instability and computational complexity of the matrix inversion of the graph Laplacians $L_A, L_B$, we make the following two changes. First, we replace the unnormalized graph Laplacian matrices $L_A$ and $L_B$ with the normalized random walk Laplacians

$$P_A = D_A^{-1} W_A \qquad P_B = D_B^{-1} W_B,$$

where $D_A, D_B$ are the degree matrices of $G_A, G_B$, respectively. The random-walk Laplacian is the basis of diffusion maps, which have been shown to be connected to spectral clustering in [29, 30]. Let $U_B \in \mathbb{R}^{p \times d_B}$ be a matrix containing the $d_B$ leading right eigenvectors of $P_B$. These vectors, scaled by their corresponding eigenvalues, are the diffusion vectors of the graph $G_B$ [9]. A vector containing the $i$-th element of the columns of $U_A$ is defined as the *diffusion map representation* of the $i$-th node.

The second change we make is to replace the inverse operation with the following projection operator,

$$Q_A = I - U_A (U_A^T U_A)^{-1} U_A^T, \qquad Q_B = I - U_B (U_B^T U_B)^{-1} U_B^T. \tag{6}$$

The matrices $Q_A, Q_B$ are projection operators to the subspace complementary to the diffusion vectors in $U_A, U_B$.

We compute the *differential vectors* of $G_A$, denoted $(V_A)_i$ and their significance level $(\sigma_A)_i$ by solving the following optimization problem

$$(\sigma_A)_i = \min_{\dim E = n-i+1} \max_{v \in S(E)} ||P_A Q_B v||_2, \tag{7}$$

where $E$ is a subspace in $\mathbb{R}^p$, $S(E)$ denotes unit Euclidean sphere in $E$, $(\sigma_A)_i$ is the optimal value of (7) and $(V_A)_i$ is the vector $v$ at which optimal value of (7) is attained. Note that the operator $P_A Q_B$ whose singular vectors are used to attain differential features is not symmetric in $A$ and $B$. The differential vectors in $V_A$ highlight components that are significant in $A$ but not in $B$. Similarly, the differential vectors $V_B$ that are equal to the singular vectors of $P_B Q_A$ highlight components that are significant in $B$ but not in $A$.

The relation between the diffusion vectors $U_A$ and the differentiating vectors $V_A$ is similar in nature to the relation between the outcome of the min-cut criterion in Eq. (3) and the double objective in Eq. (4). The diffusion vectors in $U_B$ capture significant processes that underlie state $B$. However, we are only interested in processes that differentiate between state $A$ and $B$. These are highlighted in the differential vectors $V_B$. In the experimental results in Sec. 4, we compare the diffusion maps representation to the differential vectors to illustrate this difference.

**Differential vectors for downstream analysis**  The differential vectors $V_A, V_B$ and their scores $\sigma_A, \sigma_B$ can be used in several ways. Here, and in the experimental section, we use the vectors in one of two applications: (i) Detecting of differential subsets of features. This can be done by performing $k$-means clustering over the rows of the matrices $V_A, V_B$; (ii) Computing differential meta-features via $V_A^T X_A$ and $V_B^T X_B$. In section 4 and supplementary material we provide several examples for using both options in downstream analysis (e.g., clustering, classification). In the next section we derive a theoretical justification for application (i) that is motivated by the stochastic block model. Algorithm 1 summarizes the steps of the DiSC algorithm. Further details about the computation of the graph and discussion on choice of hyperparameters is given in App. A.

---

**Algorithm 1** DiSC

---

**Input:**  Datasets $X^A$ and $X^B$
  Two kernel functions $K_A(\cdot, \cdot)$ and $K_B(\cdot, \cdot)$
**Output:** Subsets of differentiating features $V_A, V_B$
  1: Compute two graphs $G_A$ and $G_B$ on the columns of $X^A$ and $X^B$ with weights given by (1).
  2: Compute the random walk matrix, $P_A = D_A^{-1} W_A, P_B = D_B^{-1} W_B$.
  3: Calculate $U_A, U_B$, the leading right eigenvectors of $P_A, P_B$.
  4: Compute the projection matrices $Q_A, Q_B$ via Eq. (6).
  5: Compute differential vectors $V_A$ and $V_B$ via Eq. (7).
  6: Compute significance levels $\sigma_A$ and $\sigma_B$ via Eq. (7).
  7: optional: Perform k-means over the rows of $V_A$, and $V_B$.

---

**Computational Complexity**  Naively computing a graph based on a kernel function is of order $O(p^2)$. For some cases, this can be prohibitively large. However, there are simple ways to reduce the complexity to be close to linear. For example, a common approach is to compute the kernel function only for the closest k-nearest neighbors, and not for all the $O(p^2)$ pairs of features. Computing $k$-nearest neighbors for all features can be done effectively with structures such as KD-trees, with an average complexity (for each feature) of $k \log(p)$. Thus the total complexity of computing such a graph is $O(kp log(p))$. More efficient graph construction is possible with approximate nearest neighbors. A second advantage of this approach is that the Laplacian matrix is sparse, with only $kn$ non zero elements. Thus, the computation of the eigenvectors can be done efficiently as well with complexity $O(kp)$.

## 3.3  Two stochastic block models

The stochastic block model (SBM) has received a lot of attention due to its significant role in obtaining theoretical guarantees for community detection. In this setting, the individual community members are modeled as nodes in a random graph $G$. The edge weights $W_{ij}$ are sampled according to a

Bernoulli distribution, with

$$\Pr(W_{ij} = 1) = \begin{cases} p & \text{if } i, j \text{ belong to the same community} \\ q & \text{otherwise.} \end{cases} \tag{8}$$

Usually, one assumes $p > q$ such that the connectivity within a block is stronger than the connectivity between blocks, See [1] for further details.

In our setting, we model the features in both states by two random graphs $G_A$ and $G_B$. We consider a toy problem in which the features that differentiate two states $A$ and $B$ have different graph connectivity in $G_A$ and $G_B$. In graph $G_A$ the nodes are partitioned into two communities of sizes $l$ and $l + s$, with $s < l$. In contrast, the graph $G_B$ has three communities, denoted $\alpha, \beta$ and $\gamma$. Community $\alpha$ is equal to the first community in $G_A$, while $\beta, \gamma$ are a partition of the second into two groups of size $l$ and $s$.

Our goal is to detect the elements in $\gamma$ via a spectral approach whose steps are similar to Algorithm 1. For simplicity of exposition, instead of using the graph Laplacian matrix as in step 2 of the algorithm, we use the symmetric weight matrices $W_A, W_B$. Our goal in this analysis is to provide insight into our ability to recover differentiating groups such as $\gamma$ via Algorithm 1. Our main parameters of interest are the size of the differentiating group $\gamma$, and the ratio between the size of $\gamma$ and the size of the original block. Elements that depend on other parameters of the model (e.g. $p$ and $q$) are referred to as constants. Our derivation consists of three main steps. The proofs of our results are given in Appendix B. In addition, appendix E.6 presents numerical results that validate the bound in lemma 2 and test its optimality. We note that all matrix norms (i.e. $\|X\|$) in the following section and relevant appendices are the spectral norm.

**Step 1:** Here we address the non-random matrix $\mathbb{E}[W_B]$ where $\mathbb{E}[\cdot]$ is the expectation operator. Let $v_\alpha, v_\beta, v_\gamma$ be the leading three eigenvectors of $\mathbb{E}[W_B]$ and by $e_\gamma$ a binary indicator vector with elements $(e_\gamma)_i = 1$ if $i \in \gamma$.

**Lemma 1.** *The distance between $v_\gamma$ and $\frac{1}{\sqrt{s}} e_\gamma$ is bounded by*

$$\left\| v_\gamma - \frac{1}{\sqrt{s}} e_\gamma \right\| \leq C_1(p, q) \sqrt{\frac{s}{l}}.$$

*The eigenvalue corresponding to $v_\gamma$ is larger than $(p - q)s$.*

Lemma 1 shows that if the ratio $s/l$ is small enough, one can recover the elements in $\gamma$ by applying a threshold to $v_\gamma$. However, in practice we only have access to $W_B$ and $W_A$. In the next step we bound the difference between $v_\gamma$ and the eigenvector we compute by our spectral approach.

**Step 2:** Let $u_\alpha, u_\beta$ be the two leading eigenvectors of the random weight matrix $W_A$, and let $Q_{W_A} = I - u_\alpha u_\alpha^T - u_\beta u_\beta^T$. In this step we address the leading eigenvector of $Q_{W_A} W_B Q_{W_A}$.

**Lemma 2.** *Let $\tilde{v}_\gamma$ denote the leading eigenvector of $Q_{W_A} W_B Q_{W_A}$. Then,*

$$\|\tilde{v}_\gamma - v_\gamma\| \leq C_2(p, q) \frac{\sqrt{l}}{s} + C_3(p, q) \sqrt{\frac{s}{l}}. \qquad w.p \qquad 1 - \exp(-l).$$

**Step 3:** Observing the two lemmas, we see that there is a tradeoff concerning the size $s$ of the differentiating group $\gamma$. On the one hand, Lemma 1 shows that having a small value for $s$ makes the element more distinguishable in $v_\gamma$ and thus easier to detect. On the other hand, if $s < \sqrt{l}$ then the computed vector $\tilde{v}_\gamma$ might be too noisy to actually detect the relevant features.

Combining the two lemmas, we conclude with the following theorem.

**Theorem 1.** *We assume that $s, l$ are large s.t. $s, l \gg \max_i(C_i)$, and $s = l^\alpha$ with $0.5 < \alpha < 1$. We apply a threshold to $\tilde{v}_\gamma$ to determine the elements of $\gamma$. The relative number of errors $N_\epsilon/s$ is bounded by*

$$\begin{cases} C_2(p, q) l^{1-2\alpha} & 0.5 < \alpha \leq 2/3 \\ C_3(p, q) l^{\alpha-1} & 2/3 < \alpha < 1 \end{cases}. \qquad w.p \qquad 1 - \exp(-l).$$

*Proof of Theorem 1.* Let $C_4(p, q) = C_1(p, q) + C_3(p, q)$. combining Lemmas 1, 2 with the triangle inequality, and assuming $s = l^\alpha$ yields the inequality

$$\left\| \tilde{v}_\gamma - \frac{1}{\sqrt{s}} e_\gamma \right\|^2 \leq C_2(p, q)^2 l^{(1-2\alpha)} + C_4(p, q)^2 l^{(\alpha-1)} + C_2(p, q) C_4(p, q) l^{-\alpha/2}.$$

For $0.5 < \alpha < 2/3$, the dominant term is $C_2(p,q)^2 l^{1-2\alpha}$. We recover the elements in $\gamma$ by setting a threshold of $\frac{1}{2\sqrt{s}}$ to $\tilde{v}_\gamma$. Each misclassified element contributes at least $1/(4s)$ to the squared $l_2$ distance between $\frac{1}{\sqrt{s}}e_\gamma$ and $\tilde{v}_\gamma$. Thus, for $0.5 < \alpha \leq 2/3$ the relative number of errors $N_\epsilon/s$ is bounded with high probability by

$$\frac{N_\epsilon}{s} \leq \left\| \tilde{v}_\gamma - \frac{1}{\sqrt{s}}e_\gamma \right\|^2 \times (4s)/s \leq 4C_2(p,q)^2 l^{1-2\alpha}$$

A similar derivation can be done for $2/3 < \alpha < 1$. $\qquad\square$

## 4 Experiments [1]

### 4.1 Toy problems

We demonstrate the usefulness of our methodology in three toy datasets which present different scenarios of feature correlation patterns: 1) a subset of features that is uncorrelated in one dataset becomes correlated in the second 2) a subset of features that are correlated in one dataset are divided into two correlated groups in the second. 3) The third problem demonstrates generalizing DiSC to more than two datasets. In all experiments the kernels $K_A, K_B$ are the RBF kernel with an adaptive bandwidth. The choice of the all the hyperparameters (bandwidth, $d_A$ and $d_B$) are discussed in Appendix A.

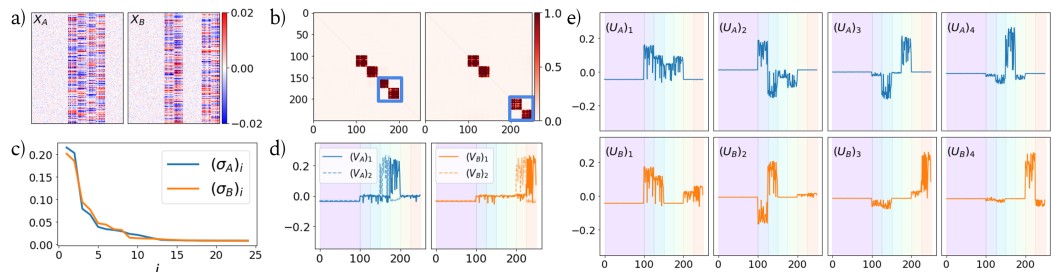

Figure 2: a) Random samples from $X^A$ and $X^B$. b) Feature correlation matrices of $X^A$ (left) and $X^B$ (right). c) Significance of the differential vectors of $X^A$ (blue) and $X^B$ (orange). d) Top two differential vectors. e) Top four diffusion maps of the features of $X^A$ (blue) and $X^B$ (orange).

**Identifying newly connected features**  We generated two datasets $X^A$ and $X^B$ with $p = 250$ features and $n_A = n_B = 10,000$ samples from a Gaussian mixture model (Fig. 2(a)), whose feature correlation is shown in Fig. 2(b). In the two datasets, the first 100 features are i.i.d samples from a normal distribution and the next 50 features are sampled from two Gaussian distributions with low rank covariance matrices. Features 151-200 in $X^A$ are sampled from two other Gaussian distributions with low rank covariance matrices, whereas in $X^B$, they are independent noise. Features 201-250 in $X^A$ are independent noise, whereas in $X^B$, they are sampled from two other Gaussian distributions with low rank covariance matrices.

Our goal is to identify the features 151-200 as the features that distinguish $X^A$ from $X^B$, and vice versa for features 201-250. We apply DiSC with $d_A = d_B = 20$. The first two significance values of the differential features are high and then significance drop drastically (Fig. 2(c)), indicating that the first two differential vectors are important in each dataset. These top two differential features for $X^A$ and $X^B$ are shown in Figure 2(d). Clearly, $(V_A)_1$ and $(V_A)_2$ highlight the features between 151-200 and more precisely, the two Gaussian mixtures are separately represented in each of these vectors. Similarly, $(V_B)_1$ and $(V_B)_2$ highlight the features between 201-250 and these represent the other two Gaussian mixtures. On the other hand, the diffusion maps eigenvectors, $U_A$ and $U_B$, captures all the connected components and not just the differential features as unique groups, as shown in Fig. 2(e).

**Identifying subsets of connected features**  We generated two datasets $X^A$ and $X^B$ with $p = 200$ features and $n_A = n_B = 10,000$ with correlation between the features as shown in Fig. 3(a). In both

---

[1] Code to reproduce the results for Sections 4.1 and 4.2 is available in `https://github.com/Mishne-Lab/DiSC`

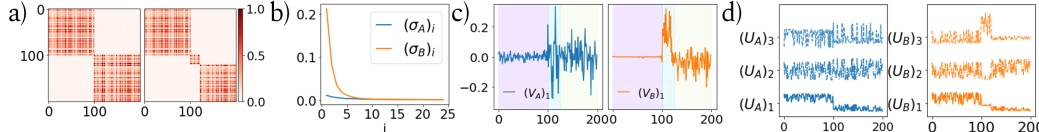

Figure 3: a) Correlation matrices. b) Significance levels. c) Differential features. d) Diffusion maps.

the datasets, the first 100 features are correlated. In $X^A$, all the remaining features form a second correlated group, yielding two connected components in the feature space. In $X^B$ the remaining features are composed of two groups of correlated features, namely, feature indices 101-125 and 126-200.

Here the goal is to identify these smaller subsets of correlated features in $X^B$, as this signifies an increase in the dimensionality of correlation structure compared to $X^A$. This also means that $X^B$ has richer feature information than $X^A$, therefore we have to identify that there are no differential features in $X^A$.

We use the significance level associated with the differential vectors to determine if the differential vectors are meaningful. We apply DiSC with $d_A = d_B = 20$. Fig. 3(c) shows the differential vectors of $X^A$ and $X^B$ and the corresponding significance levels are shown in Fig. 3(b). The significance level associated with the differential vectors of $X^A$ is negligible compared to that of $X^B$, indicating that feature information in $X^A$ is contained in $X^B$. In addition, the difference between the significance of the first two differential vectors of $X^B$ is large. Therefore, the first differential vector of $X^B$ captures the major difference, and it groups feature indices 101-125 and 126-200 with positive and negative values respectively (Fig. 3(c)). In contrast, diffusion maps captures the two connected components in $X^A$ and three connected components in $X^B$, and not just the differential components (Fig. 3(d)).

## 4.2 MNIST

The MNIST dataset [23] consists of images of hand written digits from 0 to 9, each of dimension $28 \times 28$ pixels. It has 60k training samples and 10k testing samples. We used Algorithm 2, DiSC on multiple datasets, on the MNIST dataset to extract differential features for each of the digits: features present in a single digit but are not present in each of the other digits. To have a quantitative metric to measure the significance of these differential features, we develop a classifier for a 10-class classification problem. For this, we use K-means clustering on the differential features for each digit (K=10) thus deriving clusters of pixels that together differentiate between digits. We then compute a meta-feature by averaging pixel values for each cluster. Logistic regression is performed on these meta features to distinguish between the 10 classes.

We compare our approach with Diffusion maps, Elastic Net (EN) and Elastic Net - logistic. For diffusion maps, we replaced the differential features with diffusion maps and followed the same procedure. For EN and EN-logistic, we obtained the feature importance vectors by training a classifier to distinguish between the 10-classes. K-means clustering is performed on these feature importance vectors. This cluster information is further used to compute meta-feature and finally for classification. Additionally, the entire data (784 features) is used to obtain the clusters using K-means clustering which are further used to compute classification accuracy following the procedure mentioned above. We consider this as a baseline. The results are tabulated below in Table 1.

We can see that our proposed method has the best performance, as it can capture the important differential information needed for classification. Since diffusion maps capture both shared and differential features, and entire data has all the information, their performance is poor compared to DiSC. Elastic Net has the least accuracy because it is designed for a regression setup but this is a classification problem. Elastic Net-logistic has better performance over Elastic Net as we use a cross entropy loss function – the one used for classification setup. Further, DiSC has better performance than Entire Data as it is designed to capture the differential features. However, DiSC outperformed EN-logistic as it is designed to extract groups of differential features unlike EN-logistic which just extracts all the differential features as a single group, especially when those features have similar effect on the classification problem. Note in App. E we include an additional experiment on pairwise

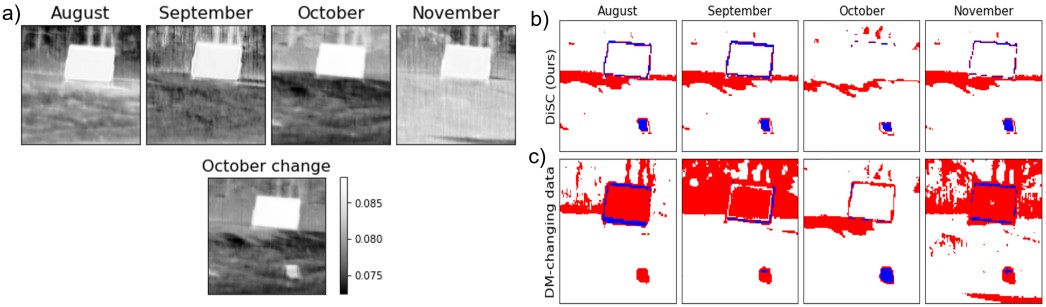

Figure 4: Hyperspectral imaging averaged across channels (a). Three clusters of features (white, red and blue) formed from DiSC (b) and Diffusion Maps for changing data (c).

classification of MNIST digits which also includes more details about the choice of hyper-parameters for each of these methods.

| | Entire Data | Diffusion Maps | EN | EN-logistic | DiSC (Ours) |
|---|---|---|---|---|---|
| Test Accuracy(%) | 76.23 | 87.42 | 69.96 | 82.42 | **89.75** |

Table 1: Classification performance on MNIST using DiSC and other competitive approaches.

### 4.3 Hyperspectral imagery

We apply DiSC to a hyperspectral imagery change detection dataset [12], consisting of hyperspectral images of a particular scene that were captured during different weather conditions, lighting conditions, and across four different months of the year (August, September, October and November). Each hyperspectral image is captured at a resolution of $100 \times 100$ pixels and 124 different bands. We denote these by $X^{\text{aug}}, X^{\text{sep}}, X^{\text{oct}}$ and $X^{\text{nov}}$. These images consist of a metal frame, grass and trees in the background, as shown in Fig. 4. An additional hyperspectral image, referred to as 'October-change' and denoted $X^{\text{oct-c}}$, was captured in October, in which an added object, a *tarp*, was included in the scene, see Fig. 4. The goal is to detect the tarp as an added object, albeit various other aforementioned different conditions during which the image is captured.

We consider $X^A \in \{X^{\text{aug}}, X^{\text{sep}}, X^{\text{oct}}, X^{\text{nov}}\}$ and $X^B = X^{\text{oct-c}}$ and compute the differential vectors for these four pairs of datasets. Note that this is similar to the toy problem presented in Sec. 4.1 and the theoretical analysis in Sec. 3.3, where a group of correlated features in $X^A$ (the pixels belonging to the grass) are divided into two groups of correlated features in $X^B$ (grass and tarp). With the addition of the tarp, the pixels belonging to the tarp remain correlated but their connectivity with the other pixels in the grass is lost. The differential features of $X^B$ identify the tarp, see App. E Fig. 8.

We pick the top four significant differential vectors of $X^B$ and perform k-means clustering on these with k=3. Figure 4(b) shows these three feature clusters for the four pairs of datasets. The tarp is revealed as a dominant cluster in all months. We compare our results with Diffusion maps for changing data (DM-changing data) [8], a spectral approach designed to capture differences between two conditions, which introduces a distance metric that measures the distance between diffusion maps calculated on a dataset that changes over time. We compute this pixel-wise distance between $X^A$ and $X^B$ and cluster the pixels into 3 groups based on this distance. These clusters are illustrated in Figure.4(c). This approach is much more affected by the acquisition conditions, For example: weather and lighting, than our approach, as it groups additional objects that have not changed along with the tarp. Finally, DiSC performs better than DM-changing data in the presence of added noise (see Appendix E).

### 4.4 fMRI

We assess the performance of DiSC in identifying groups of brain parcels with correlated BOLD activity in a working memory task from the Human Connectome Project [39], where subjects executed interleaved blocks of 0-back and 2-back working memory tasks. In these tasks, subjects are instructed to monitor a sequence of visual items and to respond whenever a presented item is the same as the one previously presented 2 items ago (2-back) or a predetermined item (0-back). This fMRI

dataset consists of 515 subjects (subjects with high motion or incomplete data were removed), and a whole-brain, functional atlas [34] was used to extract time-courses from $p = 268$ brain parcels. Dataset $X^A$ is composed of all blocks from the 0-back task and $X^B$ is composed of all blocks from the 2-back task.

For each subject we calculated the top 2 diffusion maps eigenvectors as well as the top 2 differential features for each dataset, and average these across all subjects. Fig. 5 displays the correlation of each of the averaged vectors with indicator vectors for 10 canonical brain networks [15]. Results show that diffusion maps is mainly correlated with visual networks, while the differential vector for the 0-back task being most correlated with the visual II network. On the other hand, the differential vector for the 2-back task is most correlated with the frontal-parietal network, which has been shown to be predictive of working memory performance [2]. Thus, as opposed to diffusion maps, DiSC reveals that the 2-back task incorporates more high-level cognitive regions (e.g., prefrontal) compared to the 0-back task which has lower cognitive load [18, 19].

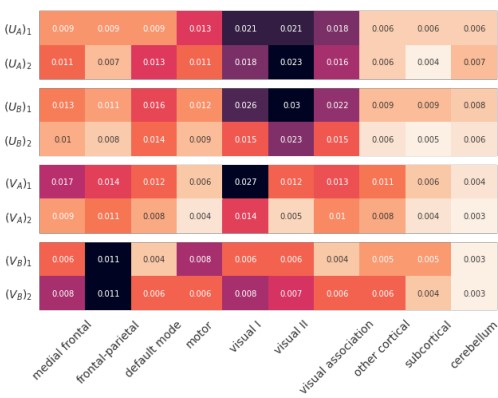

Figure 5: Canonical correlations of subject-averaged diffusion maps and differential features with 10 canonical brain networks for fMRI data of a 0-back (task $A$) and 2-back (task $B$) working memory tasks.

## 5 Discussion and future work

In this paper we introduced DiSC, a spectral approach for finding differential features between two or more datasets. We demonstrate the results of our model on various synthetic and real-world datasets and show that DiSC extracts better differential features as compared to the competing techniques. We also show the experimental results on more than two datasets. One limitation of our method is that it addresses only differences in "connectivity", or correlation, between features, not the feature values themselves. Another limitation is the choice of the hyperparameters, $d_A$ and $d_B$. High values would result in extracting noise or nuisance features as the differential features, and low values might not detect the essential differential features. Finally, the problem of "redundant" eigenvectors [11] arising in spectral clustering and manifold learning may further complicate choosing the correct dimensionality. This can be mitigated by using non-redundant eigenvectors [3] which we will explore in future work.

## 6 Acknowledgements

This research was partially supported by a Simons Foundation Pilot Award (876513SPI) and NSF award 2217058. fMRI data were provided by the Human Connectome Project, WU-MinnConsortium (Principal Investigators: David Van Essen and Kamil Ugurbil;1U54MH091657) funded by the 16 NIH Institutes and Centers that support the NIH Blueprint for Neuroscience Research. The authors thank Siyuan Gao for preprocessing of the fMRI dataset and for valuable discussions.

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
