# A   Choice of hyperparameters

The Disc algorithm requires the following hyperparameters: (i) The bandwidth for the kernel functions $K_A(X^A_{\cdot i}, X^A_{\cdot j})$, $K_B(X^B_{\cdot i}, X^B_{\cdot j})$, see Eq. (1), and (ii) The number of significant eigenvectors computed for $G_A$ and $G_B$, denoted $d_A$ and $d_B$, respectively.

**Self-tuning bandwidth**   For computing the weight matrices, we use the self-tuning bandwidth from [47] where the bandwidth for an RBF kernel is given by $K(x_i, x_j) = \exp(\|x_i - x_j\|/\sigma_i\sigma_j)$. The local bandwidth $\sigma_i$ for each node is set to the distance to its $k$-th nearest neighbor, as suggested in [47] and as is common in practice. The rule of thumb for choosing $k$ is around $\log(p)$ where $p$ is the number of features.

**Determining $d_A$ and $d_B$**   The notation $d_A, d_B$ represent the number of significant eigenvectors present in the random walk matrices of $P_A, P_B$ respectively. It is important to note that if $d_A, d_B$ are two small, the leading singular vectors of $P_A Q_B$ and $P_B Q_A$ will include elements of the shared latent space. The results will not change dramatically, however, if the choice of $d_A, d_B$ is higher than the optimum. For example, in the experiment in section 4.1 "Identifying newly connected features", when we consider $d_A = d_B < 4$, the shared latent space of features between 100-150 are also highlighted in $V_A, V_B$ which is undesirable. However, we can increase the values of $d_A, d_B$ up to around 150, with very little impact on the results. For the MNIST data, we computed differential features between digits 4 and 9 with various values for $d_A = d_B = d$ and followed the procedure mentioned in the paper to compute the classification accuracy. These results are given in Table 2. We can see that for very small values of $d$, the accuracy is lower since the classifier is partially trained on information about shared features. Here as well, there is a wide range of values (between 20-40) that yield similar results.

| $d_A = d_B = d$ | 10 | 20 | 30 | 40 | 50 | 60 |
|---|---|---|---|---|---|---|
| Test Accuracy | 95.2% | 96.5 % | 96.5% | 96.4% | 94.7 % | 88.5% |

Table 2: Impact of the choice of hyperparameter $d$ on the classifier accuracy for pairs of MNIST digits.

# B   Proof of lemmas 1 and 2

## B.1   Preliminaries

**The Davis-Kahan Theorem**   In our proof, we make a repeated use of the Davis-Kahan theorem. We apply both the classic theorem, and a useful variant derived in [45].

**Theorem 2** ([45], Theorems 1 and 2). *Let $W$ be a symmetric matrix with eigenvectors $v_1, \ldots, v_n$ and corresponding eigenvalues $\lambda_1 \geq \lambda_2, \ldots, \lambda_n$. We denote by $\widetilde{W}$ a perturbation of $W$, with eigenvectors $\tilde{v}_1, \ldots, \tilde{v}_n$ and corresponding eigenvalues $\tilde{\lambda}_1 \geq \tilde{\lambda}_2, \ldots, \tilde{\lambda}_n$. Let $T_W, T_{\widetilde{W}}$ be the projection matrices onto a subspace spanned by the leading $d$ eigenvectors,*

$$T_W = \sum_{i=1}^{d} v_i v_i^T \qquad T_{\widetilde{W}} = \sum_{i=1}^{d} \tilde{v}_i \tilde{v}_i^T.$$

*In addition, we define $\delta$ via,*

$$\delta = \min_{j \leq d; i > d} |\tilde{\lambda}_i - \lambda_j|.$$

*Then,*

$$\|T_W - T_{\widetilde{W}}\| \leq \|W - \widetilde{W}\|/\delta.$$

*Alternatively,*

$$\|T_W - T_{\widetilde{W}}\| \leq 2\sqrt{d}\|W - \widetilde{W}\|/(\lambda_k - \lambda_{k+1}).$$

The first inequality is the classic Davis-Kahan theorem and the second is its variant derived in [45]. The importance of the variant is that it bounds the eigenvector perturbation as a function of the

eigenvalues of original matrix $W$, with no dependency on the eigenvalues of the perturbed matrix, which are unknown in many cases. In addition, the bound on the projection matrices $T_W - T_{\widetilde{W}}$ can be replaced with a bound on the difference in norm between the subspace of eigenvectors, see for example Corollary 3 in [45]. For a single vector we have

$$\|\tilde{v}_i - v_i\| \leq \frac{2^{3/2}\|W - \widetilde{W}\|}{\min(\lambda_{i-1} - \lambda_i, \lambda_i - \lambda_{i+1})}. \tag{9}$$

## B.2 Concentration of weight matrix for stochastic block models

Another useful result is the concentration, in spectral norm, of a weight matrix generated according to the stochastic block model. This result follows directly from Bernstein's inequality for sums of independent matrices with bounded norm. This derivation is clearly presented, for example, in [40]

**Lemma 3.** *Let $W \in \mathbb{R}^{l \times l}$ be a matrix generated by the stochastic block model as in Eq. (8). Then,*

$$\|W - \mathbb{E}[W]\| = C\sqrt{l} \qquad \text{with probability} \qquad 1 - \exp(-l).$$

The rank of the expected weight matrix $\mathbb{E}[W]$ is equal to the number of communities in the model. Assume $d$ communities and let $T_W$ and $T_{\mathbb{E}[W]}$ denote the projection matrices onto the leading $d$ eigenvectors of $W$ and $\mathbb{E}[W]$ respectively. Let $\lambda_d$ denote the $d$-th eigenvalue of $\mathbb{E}[W]$. Combining Lemma 3 and Theorem 2 yields the following perturbation bound,

$$\|T_W - T_{\mathbb{E}[W]}\| \leq \frac{C\sqrt{dl}}{\lambda_d}. \tag{10}$$

where $C$ is a constant that does not depend on the parameters of the model.

## B.3 Auxiliary lemmas

For the lemmas in this subsection we have the following notation. Let $W_A, W_B \in \mathbb{R}^{(2l+s) \times (2l+s)}$ be random weight matrices obtained via the stochastic block model as described in Section 3.3. Let $Q_{W_A}, Q_{\mathbb{E}[W_B]}$ denote two projection matrices onto the complementary subspace of the leading eigenvectors of $W_A$ and $\mathbb{E}[W_B]$ respectively.

**Lemma 4.** *We have the following bound on the numerator of Eq. (15).*

$$\|Q_{W_A} W_B Q_{W_A} - Q_{\mathbb{E}[W_B]} \mathbb{E}[W_B] Q_{\mathbb{E}[W_B]}\| \leq C_1 \sqrt{l} + C_2 \sqrt{\frac{s}{l}} \lambda_3$$

*Proof.* We denote by $\mathcal{E} = Q_{W_A} - Q_{\mathbb{E}[W_B]}$. Applying the triangle inequality and the Cauchy-Schwarts inequality,

$$
\begin{aligned}
&\|Q_{W_A} W_B Q_{W_A} - Q_{\mathbb{E}[W_B]} \mathbb{E}[W_B] Q_{\mathbb{E}[W_B]}\| \\
&= \|Q_{W_A} W_B (Q_{\mathbb{E}[W_B]} + \mathcal{E}) - (Q_{W_A} - \mathcal{E}) \mathbb{E}[W_B] Q_{\mathbb{E}[W_B]}\| \\
&\leq \|Q_{W_A} (W_B - \mathbb{E}[W_B]) Q_{\mathbb{E}[W_B]}\| + \|\mathcal{E} \mathbb{E}[W_B] Q_{\mathbb{E}[W_B]}\| + \|Q_{W_A} W_B \mathcal{E}\|. \\
&\leq \|W_B - \mathbb{E}[W_B]\| + \|\mathcal{E}\| \|\mathbb{E}[W_B] Q_{\mathbb{E}[W_B]}\| + \|Q_{W_A} W_B\| \|\mathcal{E}\|.
\end{aligned} \tag{11}
$$

The term $\|\mathbb{E}[W_B] Q_{\mathbb{E}[W_B]}\|$ is equal by definition to the third eigenvalue of $\mathbb{E}[W_B]$. We combine lemmas 3, 5 and 6 to get,

$$\|Q_{W_A} W_B Q_{W_A} - Q_{\mathbb{E}[W_B]} \mathbb{E}[W_B] Q_{\mathbb{E}[W_B]}\| \leq C\sqrt{l} + \sqrt{\frac{s}{l}} \lambda_3 + C_2 \sqrt{s}.$$

$\square$

**Lemma 5.** *The error of $\|\mathcal{E}\| = \|Q_{W_A} - Q_{\mathbb{E}[W_B]}\|$ is bounded by,*

$$\|\mathcal{E}\| \leq C\sqrt{\frac{s}{l}}.$$

*Proof.* The triangle inequality yields,

$$\|Q_{W_A} - Q_{\mathbb{E}[W_B]}\| \leq \|Q_{W_A} - Q_{\mathbb{E}[W_A]}\| + \|Q_{\mathbb{E}[W_A]} - Q_{\mathbb{E}[W_B]}\|$$

For the first term, we use the results for the standard stochastic block model of sizes $l$ and $l + s$. The second (and smallest non-zero) eigenvalue of $\mathbb{E}[W_A]$ is larger than $(p - q)\frac{l}{2}$

$$\|Q_{W_A} - Q_{\mathbb{E}[W_A]}\| \leq \frac{C\sqrt{l}}{(p - q)l} = \frac{C}{\sqrt{l}(p - q)}.$$

The second term bounds the difference in projection matrices of the subspace spanned by the leading two eigenvectors of $\mathbb{E}[W_A]$ and $\mathbb{E}[W_B]$. In the proof of lemma 1 we show that the difference between the leading eigenvectors of the two matrices is bounded by $\sqrt{\frac{s}{l}}$ and that the second eigenvector is identical. It follows that,

$$\|Q_{W_A} - Q_{\mathbb{E}[W_B]}\| \leq \frac{C}{\sqrt{l}(p - q)} + \sqrt{\frac{s}{l}}.$$

$\square$

**Lemma 6.** *The value of $\|Q_{W_A} W_B\|$ is bounded by,*

$$\|Q_{W_A} W_B\| \leq C\sqrt{l}.$$

*Proof.* We use the triangle inequality and Cauchy Schwartz to split the term into the following,

$$
\begin{aligned}
\|Q_{W_A} W_B\| &\leq \|Q_{W_A} W_A\| + \|Q_{W_A}(W_A - W_B)\| \leq \|Q_{W_A} W_A\| + \|Q_{W_A}(\mathbb{E}[W_A] - \mathbb{E}[W_B])\| \\
&\quad + \|Q_{W_A}(W_B - \mathbb{E}[W_B])\| + \|Q_{W_A}(W_A - \mathbb{E}[W_A])\| \quad\quad\quad (12)\\
&\leq \|Q_{W_A} W_A\| + \|W_B - \mathbb{E}[W_B]\| + \|W_A - \mathbb{E}[W_A]\| + \|Q_{W_A}(\mathbb{E}[W_A] - \mathbb{E}[W_B])\| \\
&\leq \|Q_{W_A} W_A\| + \|W_B - \mathbb{E}[W_B]\| + \|W_A - \mathbb{E}[W_A]\| + \|Q_{\mathbb{E}[W_A]}(\mathbb{E}[W_A] - \mathbb{E}[W_B])\| \\
&\quad + \|\mathbb{E}[W_A] - \mathbb{E}[W_B]\|\|Q_{\mathbb{E}[W_A]} - Q_{W_A}\|
\end{aligned}
$$

By lemma 3 (concentration of the norm of random matrix) The terms $\|W_B - \mathbb{E}[W_B]\|$ and $\|W_A - \mathbb{E}[W_A]\|$ are bounded by $C\sqrt{l}$. The term $\|Q_{W_A} W_A\|$ is equal by definition to the third eigenvalue of $W_A$. Recall that the third eigenvalue of $\mathbb{E}[W_A]$ is equal to zero. Thus, by weyl's inequality, the third eigenvalue of $W_A$ is bounded by

$$\lambda_3(W_A) \leq \|W_A - \mathbb{E}[W_A]\| \leq C\sqrt{l}.$$

To bound the term $\|Q_{WA}(\mathbb{E}[W_A] - \mathbb{E}[W_B])\|$ note that the matrix $\mathbb{E}[W_A] - \mathbb{E}[W_B]$ is deterministic, and equal to

$$\mathbb{E}[W_A] - \mathbb{E}[W_B] = (p - q)e_\beta e_\gamma^T$$

with a norm of $(p - q)\sqrt{sl}$. The operator $Q_{W_A}$ removes the average from the vector $v_\beta$ over the $l + s$ last elements. Thus, the norm of $\|Q_{\mathbb{E}[W_A]}e_\beta\|$ is bounded by $s/l$, which implies that

$$\|Q_{\mathbb{E}[W_A]}(\mathbb{E}[W_A] - \mathbb{E}[W_B])\| \leq (p - q)s^{3/2}/l.$$

Finally, the last term is bounded by $C\frac{\sqrt{sl}}{(p-q)\sqrt{l}} = C\frac{\sqrt{s}}{p-q}$, Which is dominated by the first four terms. Summing up the different terms we have,

$$\|Q_{W_A} W_B\| \leq C\sqrt{l} + (p - q)s^{3/2}/l.$$

$\square$

### B.4 Proof of lemma 1

*Proof.* We define $e_\alpha$, $e_\beta$ and $e_\gamma$ as the binary block indicator vectors where $(e_\alpha)_i = 1$ if $i \in \alpha$, $(e_\beta)_i = 1$ if $i \in \beta$ and $(e_\gamma)_i = 1$ if $i \in \gamma$. Let $E \in \mathbb{R}^{(2l+s)\times 3}$ be a concatenation of $e_\alpha, e_\beta$ and $e_\gamma$. We denote the pairwise block confusion matrix $\Theta \in [0, 1]^{3\times 3}$ given by,

$$\Theta_{ij} = \begin{cases} p & i = j \\ q & i \neq j. \end{cases} \quad\quad\quad (13)$$

The expected weight matrix of the stochastic block model is equal to,

$$\mathbb{E}[W_B] = E\Theta E^T.$$

We denote by $\Delta \in R^{3\times 3}$ a diagonal matrix with,

$$\Delta_{11} = \sqrt{l} \quad \Delta_{22} = \sqrt{l} \quad \Delta_{33} = \sqrt{s}.$$

The expected weight matrix of the stochastic block model is equal to

$$\mathbb{E}[W_B] = E\Theta E^T = (E\Delta^{-1})(\Delta\Theta\Delta)(\Delta^{-1}E^T).$$

The matrix $E\Delta^{-1}$ is orthonormal. The eigenvalues of $\mathbb{E}[W_B]$ are thus equal to the eigenvalues of $\Delta\Theta\Delta$ and the corresponding eigenvectors are equal to $E\Delta^{-1}$ multiplied by the eigenvectors of $\Delta\Theta\Delta$. Consider the matrix $Z$ given by,

$$Z_{ij} = \begin{cases} (\Delta\Theta\Delta)_{ij} & i,j < 3 \ \text{ or } \ i=j=3 \\ 0 & o.w. \end{cases}$$

The eigenvectors of $Z$ are equal to $u_1 = [1,1,0]$ and $u_2 = [1,-1,0]$ with corresponding eigenvalues $(p+q)l$ and $(p-q)l$ respectively. We denote by $\tilde{u}_1, \tilde{u}_2, \tilde{u}_3$ the eigenvectors of $\Delta\Theta\Delta$. A direct computation shows that $\tilde{u}_2 = u_2$ with the same eigenvalue. Applying Theorem 2 yields,

$$\|u_1 - \tilde{u}_1\| \leq \frac{2\sqrt{2}\|Z - \Delta\Theta\Delta\|}{2ql} = \frac{4\sqrt{2}q\sqrt{ls}}{2ql} = \sqrt{\frac{8s}{l}}. \tag{14}$$

Both $u_1, u_3$ and $\tilde{u}_1, \tilde{u}_3$ are orthogonal to $u_2 = \tilde{u}_2$ and thus span the same 2D subspace. This implies

$$\|\tilde{u}_3 - u_3\| = \|u_1 - \tilde{u}_1\| \leq \sqrt{\frac{8s}{l}}.$$

It follows directly that $v_\gamma$, the the third eigenvector of $\mathbb{E}[W_B]$ satisfies

$$\left\|v_\gamma - \frac{1}{\sqrt{s}}e_\gamma\right\| \leq \sqrt{\frac{8s}{l}}.$$

Finally, we derive a lower bound on the third eigenvalue of $\mathbb{E}[W_B]$. Recall that the eigenvalues of $\mathbb{E}[W_B]$ are equal to those of $\Delta\Theta\Delta$. For the latter we apply the inequality,

$$\lambda_{\min}(\Delta\Theta\Delta) \geq \lambda_{\min}(\Delta)\lambda_{\min}(\Theta)\lambda_{\min}(\Delta),$$

where $\lambda_{\min}$ denotes the smallest eigenvalue. The matrix $\Delta$ is diagonal with values $\sqrt{l}, \sqrt{l}, \sqrt{s}$ and hence $\lambda_{\min}(\Delta) = \sqrt{s}$. By direct computation $\lambda_{\min}(\Theta) \geq (p-q)$. Hence,

$$\lambda_3(\mathbb{E}[W_B]) = \lambda_{\min}(\Delta\Theta\Delta) \geq s(p-q).$$

$\square$

## B.5 Proof of Lemma 2

In this lemma we bound the difference between the vector $\tilde{v}_\gamma$ and the corresponding eigenvector of $\mathbb{E}[W_B]$, denoted by $v_\gamma$. Recall the definition of $Q_{\mathbb{E}[W_B]}$ as the projection matrix onto the complementary subspace of the two leading eigenvectors of $\mathbb{E}[W_B]$. By definition, $v_\gamma$ is the leading eigenvector of $Q_{\mathbb{E}[W_B]}\mathbb{E}[W_B]Q_{\mathbb{E}[W_B]}$. Thus, our goal is to bound the leading eigenvector of two matrices:

$$Q_{\mathbb{E}[W_B]}\mathbb{E}[W_B]Q_{\mathbb{E}[W_B]} \quad \text{and} \quad Q_{\mathbb{E}[W_A]}W_B Q_{\mathbb{E}[W_A]}.$$

To that end, we apply the Davis-Kahan theorem. Let $\lambda_\gamma$ be the third eigenvector of $\mathbb{E}[W_B]$. The theorem gives the following bound.

$$\|v_\gamma - \tilde{v}_\gamma\| \leq 2^{3/2}\frac{\|Q_{\mathbb{E}[W_B]}\mathbb{E}[W_B]Q_{\mathbb{E}[W_B]} - Q_{\mathbb{E}[W_A]}W_B Q_{\mathbb{E}[W_A]}\|}{\lambda_3}. \tag{15}$$

Applying lemma 4 we get:

$$\|v_\gamma - \tilde{v}_\gamma\| \leq C\frac{\sqrt{l}}{\lambda_3} + C_2\sqrt{\frac{s}{l}}$$

For the first term, we apply the lower bound $\lambda_3 \geq (p-q)s$. This yields,

$$\|v - \tilde{v}\| \leq C_1\frac{\sqrt{l}}{s} + C_2\sqrt{\frac{s}{l}}$$

In Section E.6 we provide a numerical validation of Lemma 4 and the bound on $\lambda_3$.

## C Alternative justification for our approach

A different variation of the graph-cut criterion in (2) is the Normalized cut. Let $d_i$ denote the degree of node $i$, and let $\text{vol}(\alpha) = \sum_i d_i$ be the sum of degrees of a subset of nodes $\alpha$. The normalized cut is equal to,

$$\text{Ncut}(\alpha, \beta) = \sum_{i \in \alpha, j \in \beta} W(i,j)\Big(\frac{1}{\text{vol}(\alpha)} + \frac{1}{\text{vol}(\beta)}\Big).$$

Similarly to (3), we can define an indicator vector by,

$$f_i = \begin{cases} \sqrt{\frac{\text{vol}(\beta)}{\text{vol}\alpha}} & i \in \alpha \\ \sqrt{\frac{\text{vol}(\alpha)}{\text{vol}\beta}} & i \in \beta. \end{cases} \tag{16}$$

The normalized cut can be expressed as,

$$\text{Ncut}(\alpha, \beta) = f^T L f.$$

Here, the indicator vector is orthogonal to $D\mathbf{1}$, where $D$ is a diagonal matrix containing the degrees $\{d_i\}$ in its diagonal and $\mathbf{1}$ is the all ones vector. In addition, though the norm of $f$ depends on $\alpha$, the term $f^T D f$ is a constant. Relaxing the requirement in Eq. (16), yields the following optimization problem for the normalized cut,

$$\min_f f^T L f \qquad \text{subject to} \qquad f^T D f = 1.$$

In our work, we consider two graph Laplacians $L_A, L_B$. For simplicity we assume that the degree matrix of both graphs is identical. Thus, the indicator vector in Eq. (16) is the same for both graphs and for any partition. The second graph $L_B$ yields a second constraint on the optimization problem.

$$\min f^T L_A f \quad \text{subject to} \quad f^T D f = 1, \quad f^T L_B f = \gamma. \tag{17}$$

The solution to Eq. (17) satisfies the generalized eigenvector problem

$$L_A f = \lambda_1 (D + \lambda_2 L_B) f.$$

Multiplying both sides by $D^{-1}$ yields,

$$(I + \lambda_2 P_b)^{-1} P_A f = \lambda_1 f, \tag{18}$$

where $P_A, P_B$ are the random walk Laplacian matrices. Note that the term $(I + \lambda_2 D^{-1} L_b)^{-1}$ is a *regularized inverse* matrix of $P_B$, often used to ovoid an arbitrary increase of noise when computing the inverse of ill condition matrices. Thus, the expression in (18) is very similar in nature to the expression in Eq. 5.

## D DiSC extension to multiple datasets

The extension of our approach to multiple datasets (i.e. more than two) is straightforward. We consider $M$ datasets $X^m \in \mathbb{R}^{n_m \times p}$ for $m = 1, .., M$. We compute random-walk transition matrices on the graphs by $P_m = D_m^{-1} W_m$. Let $U_m \in \mathbb{R}^{p \times d_m}$ be a matrix containing the $d_m$ leading right eigenvectors of $P_m$, and let $\hat{Q}_m$ be a projection matrix onto the the complementary subspace of $\cup_{k \neq m} U_k$, given by

$$\hat{Q}_m = I - \hat{U}_m (\hat{U}_m^T \hat{U}_m)^{-1} \hat{U}_m^T. \tag{19}$$

where $\hat{U}_m = [U_1, ..., U_{m-1}, U_{m+1}, ..., U_M]$. We then compute the differential vectors of $G_m$, denoted by $(V_m)_i$ and the corresponding significance level $(\sigma_m)_i$ using

$$(V_m)_i = \arg\max_{\dim E = i} \min_{v \in S(E)} ||P_m \hat{Q}_m v||_2. \tag{20}$$

$$(\sigma_m)_i = ||P_m \hat{Q}_m (V_m)_i||_2. \tag{21}$$

where $E$ is a subspace in $\mathbb{R}^p$ and $S(E)$ denotes the unit Euclidean sphere in $E$. An algorithm implementing our approach for multiple datasets is given in Alg. 2.

---

**Algorithm 2** DiSC - Extension to multiple datasets

---

**Input:**  Datasets $X^m \in \mathbb{R}^{n_m \times p}$ for $k = 1 : M$
           Kernel functions $\{K_m(\cdot, \cdot)\}_{k=1:M}$
           Hyper parameters $\{d_m\}_{k=1:M}$
**Output:** Subsets of differentiating features $\{V_m\}_{m=1:M}$

1: Compute graphs $G_m$ on the columns of $X^m$ with weights given by (1).
2: For all the graphs, compute the random walk matrix,

$$P_m = D_m^{-1} W_m \qquad \text{for } m = 1, ..., M$$

3: Calculate $\{U_m\}_{m=1:M}$, the leading right eigenvectors of $\{P_m\}_{m=1:M}$.
4: Compute the projection matrices $\{\hat{Q}_m\}_{m=1:M}$ via Eq. (19).
5: Compute differential vectors $\{V_m\}_{m=1:M}$ via Eq. (20).
6: Compute significance levels $\{\sigma_m\}_{m=1:M}$ via Eq. (21).
7: optional: Perform k-means over the rows of $V_A$, and $V_B$.

---

To demonstrate this extension, in the following example we generalize our approach to reveal differential features given three datasets $X_A, X_B$ and $X_C$. Some of the features connections are specific to each dataset and the remaining connections may be present in more than one dataset. Each dataset is consists of $p = 400$ features and $n_A = n_B = n_C = 10,000$, with feature correlations as shown in Fig. 6(a). Dataset specific groups of correlated features are highlighted. That is, feature connections between 301-350, 201-250 and 351-400 are specific to $X^A$, $X^B$ and $X^C$, respectively. Note that each dataset has two groups of correlated features that are dataset specific and the aim is to identify these two groups of features individually. We apply DiSC with $d_A = d_B = d_C = 20$. Differential features are shown in Fig. 6(c). The significance level of the first two differential vectors is almost similar for all the datasets and then significance level drops, as shown in Fig. 6(b). This indicates that the first two differential vectors capture the major differences. These vectors, shown in Fig. 6(b), clearly represent the dataset specific connections and each differential vector identifies a group of connected features, unlike diffusion maps in Fig. 6(d).

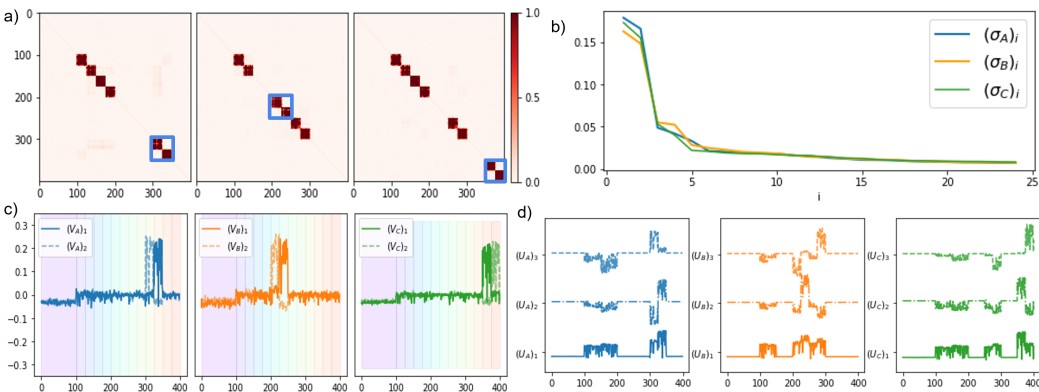

Figure 6: Multiple datasets. a) Correlation matrices. b) Significance levels. c) Differential features. d) Diffusion maps.

# E   Additional experimental results

**Identifying subsets of connected features in both datasets**    We generate a toy problem with two datasets $X^A$ and $X^B$ with $p = 200$ features and $n_A = n_B = 10,000$, whose features are correlated as in Fig. 7(a). There is a subset of correlated features in $X^A$ that are divided into two groups in $X^B$ and vice versa. $X^A$ has three groups of correlated features with feature indices [1-75],[76-100] and [101-200]. $X^B$ has another three groups of correlated features with feature indices [1-100],[101-125],[126-200]. Thus, feature indices [1-100] which have strong connectivity in $X^B$ are divided

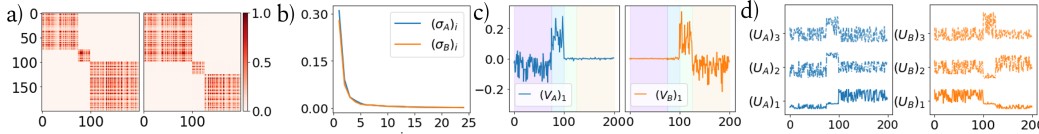

Figure 7: a) Correlation matrices. b) Significance levels c) Differential features d) Eigenvectors of diffusion maps.

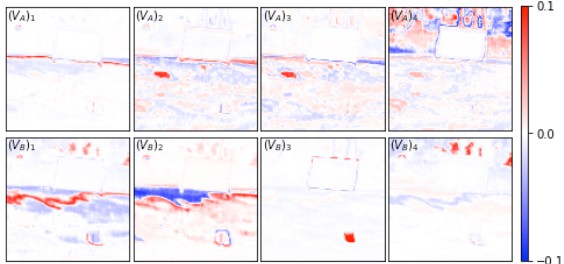

Figure 8: The differential vectors for $X^A = X^{oct}$ (top row) and $X^B = X^{oct-c}$ (bottom row).

into [1-75] and [76-100] groups in $X^A$. Similarly, feature indices [101-200] which have strong connectivity in $X^A$ are divided into [101-125] and [126-200] groups in $X^B$. The goal is to identify these sub-divided groups i.e., feature indices [1-75],[76-100] in $X^A$ and [101-125],[126-200] in $X^B$ as the differential features. Significant differential vectors and the corresponding significance values are shown in Fig. 7(c) and (b) respectively. The significance level of the differential vector in both datasets is roughly similar and the significance level drops after the first vector. Therefore, the first differential vector is the most significant one. Fig. 7(c) shows that the differential vectors are clustering the two subsets of differential features.

### E.1 Hyperspectral imaging

We provide two additional results of DiSC on the hyperspectral imaging dataset. To illustrate the feature grouping capability of DiSC, we choose $X^A = X^{oct}$ and $X^B = X^{oct-c}$, and set the hyperparameters $d_A = d_B = 20$. The top four differential features of $X^A$ and $X^B$ are shown in Fig. 8 in the top row and bottom row respectively. $(V_B)_3$ captures the tarp and $(V_B)_4$ captures the slight variation in the trees in the background. $(V_B)_1$ and $(V_B)_2$ captures the change in the grass. Clearly, there is a grouping effect and each group of differential features are captured individually. Also, since the entire information in $X^A$ is present in $X^B$, $V_A$ do not capture any differences.

To evaluate the robustness of our approach to noise, we added Gaussian noise with $\mu = 0$ and $\sigma = 0.01$ to all datasets. We then compute the differential features using DiSC and distance measure using DM-changing data. We cluster the features using these with k-means clustering and $k=3$. Fig. 9 shows the clusters for four pairs of datasets using DiSC (top row) and DM-changing data (bottom row). DiSC is more robust to noise than DM-changing data. In most months it groups the pixels belonging to the tarp as a practically separate cluster of differential features, as opposed to DM-changing data which groups the tarp with other features in the background. Thus, DiSC is more invariant to imaging conditions and highlights the object that has changed in the scene.

### E.2 MNIST pairwise classification

The MNIST dataset [23] consists of images of hand written digits from 0 to 9, each of dimension $28 \times 28$ pixels. It has 60k training samples and 10k testing samples. We extract the differential features between each pair of digits with $d_A = d_B = 20$. The top 3 differential features from the two digits are further concatenated columnwise to form a matrix of dimension $784 \times 6$. For each pair of digits, we group the features into three clusters using the k-means algorithm applied to this matrix of concatenated differential features. We compare our approach with Naive Elastic Net (EN),

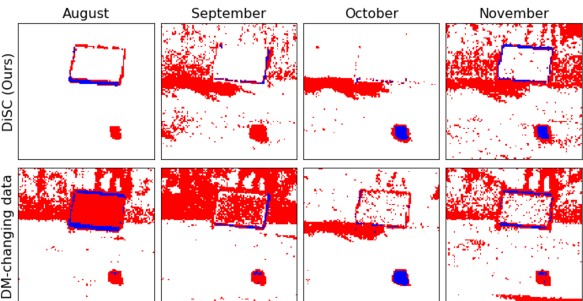

Figure 9: Noisy data. Three groups of features (white, red and blue) formed from DiSC (top row) and Diffusion Maps for changing data (bottom row).

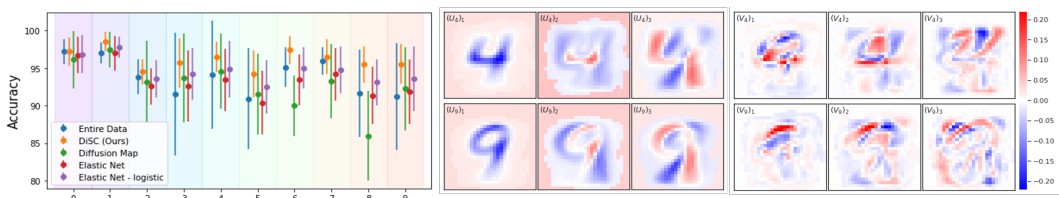

Figure 10: Results for MNIST data. The figure shows the accuracy of various methods for classifying a pair of digits. For each digit (x-axis) we show the average accuracy over all pairs.

an Elastic Net variation for classification (EN-logistics) and Diffusion Maps applied to the features of each dataset. Each of these methods yields a vector(s) representing feature importance, which is then used for feature grouping using k-means with $k = 3$. As an additional baseline, we group the features based on the entire data. To have a quantitative metric to measure the performance of these methods, we first compute the average feature values for each cluster of features and use these to train a logistic regression classifier between pairs of digits, and measure the classification accuracy on the test samples.

For each digit, Fig. 10 shows the average of its pairwise accuracy with the other digits, for different methods. Our method consistently performs better than the other methods, with relatively less variability in accuracy across different pairs of digits. The baseline model and diffusion maps have a huge variability, which may be because these are not designed to extract differences between the digits. In most cases, EN-logistic performs better than diffusion maps because EN-logistic assigns feature importance based on the classification task. However, it only has a single set of coefficients for features, whereas our method has a number of differential features with associated significance level. As illustrated in Fig. 10 for digits 4 and 9, diffusion maps captures the overall structures whereas DiSC explicitly capture their differences.

### E.3 single cell RNA sequencing

Using the *splatter* simulator [46] we generated a simulated dataset containing the RNA expression level of $p = 500$ genes, as measured in 1000 cells, which belong to two different cell types. Out of the 500 genes, only approximately 100 genes are *differential*, and thus have a different expression level for the two cell types.

We visualize the cells in 2D using t-SNE [38]. Figure 11 colors the cells by *meta features*, that are equal to weighted sum over the gene expression profile of each cell. In the upper four panels, the weights are computed by the two leading diffusion vectors of the graphs $G_A$ (left two panels) and $G_B$ (right two panels). In the bottom four panels, the weights are equal to the differential vectors $V_A$ (left two panels) and $V_B$ (right two panels). Clearly, the differential vectors highlight genes that are more relevant for differentiating between the two groups.

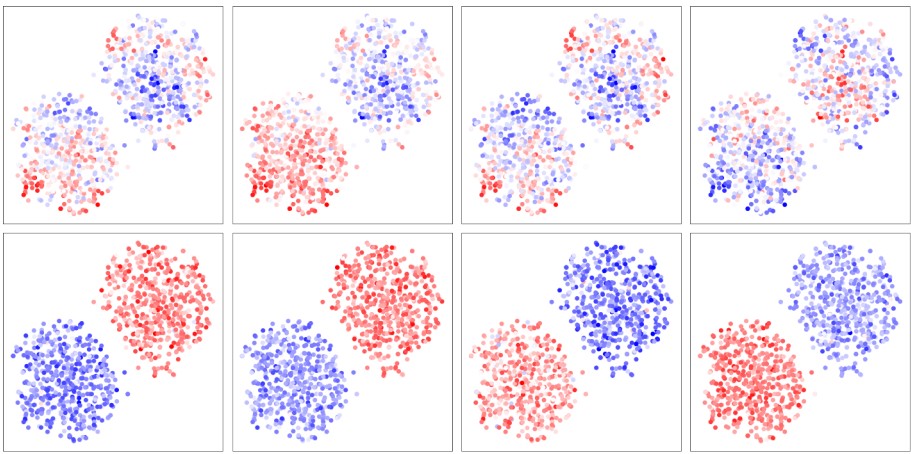

Figure 11: 2D t-SNE plots of simulated scRNA-seq generated by splatter [46]. The two clusters represent two cell types. In all panels cells are colored by *meta-features* - a weighted sum on the gene expression, corresponding to diffusion maps on each of the feature graphs separately (top row), or DiSC vectors (bottom row).

### E.4    Differential feature extraction in partially correlated conditions

We empirically analyze the performance of detecting differential features in a non-ideal case, where, say, correlated components in one dataset are partially correlated in other dataset. In section 4.1, we compared two ideal settings - features that are highly correlated in one dataset are completely independent in another. Here, we added an extension to this experiment to test non ideal cases. To that end, we introduce a parameter $p$ that determines the correlation level. Specifically, we set the eigenvalues of the covariance to decrease exponentially such that the $i^{th}$ eigenvalue $\lambda_i$ is proportional to $exp(-i * p)$. Thus, if $p$ is close to 0 - the decrease is very small, which results in a covariance matrix close to identity and thus no correlation. If p is high, then the decrease is fast which results in high correlation. In the experiment, we kept the covariance of features 151-200 in $X^A$ fixed with $p = 1$. For these features of $X^B$ we changed the values of $p$ between 0 and 1 and computed the differential vectors and significance level for each of the $p$ values. As expected the significance level of the differential vectors of $X^A$ decreases as p gets close to 1. For intermediate levels, the significance level is still high. The results are shown in Figure 12.

### E.5    Iterative baselines on simulated dataset

We applied the best subset selection approach to the simulated example in section 4.1, Identifying newly connected features. Here, our proposed methodology was able to identify features 151-200 and 201-250 as the differential features in $X^A$ and $X^B$ respectively. We applied the iterative feature selection approach to select the top 100 features that best differentiate $X^A$ and $X^B$. For our best subset selection, as a criterion, we used the accuracy of a nearest neighbor (NN) and logistic regression classifiers, that take as input the selected features. The accuracy was very poor compared to our group feature selection method, as shown in figure 13. The figure on the left (right) indicates the selected features while using NN classifier (logistic regression classifier). The selected features are indicated by value 1. Clearly, these methods do not highlight the groundtruth 151-250 features as the features that best distinguish $X^A$ and $X^B$.

### E.6    Simulations for validating Lemma 2

We ran several simulations to validate the theoretical results obtained in the proof of Lemma 2. In our simulations, the block size $l$ varied from 500 to 2000. We set $s = l^\alpha$ for various values of $\alpha$ between 0.6 and 0.9. Our goal is to obtain numerical approximation to the rate of convergence of several factors in our proof, as a function of the block size $l$. Specifically, we estimate the increase rate of the numerator and denominator of Eq. (15). The denominator is equal to $\lambda_3$, the third eigenvalue of $W_B$. By Lemma 1, $\lambda_3$ is proportional to $s = l^\alpha$. Figure 14 draws $\lambda_3$ as a function of $l$ on a log scale. The

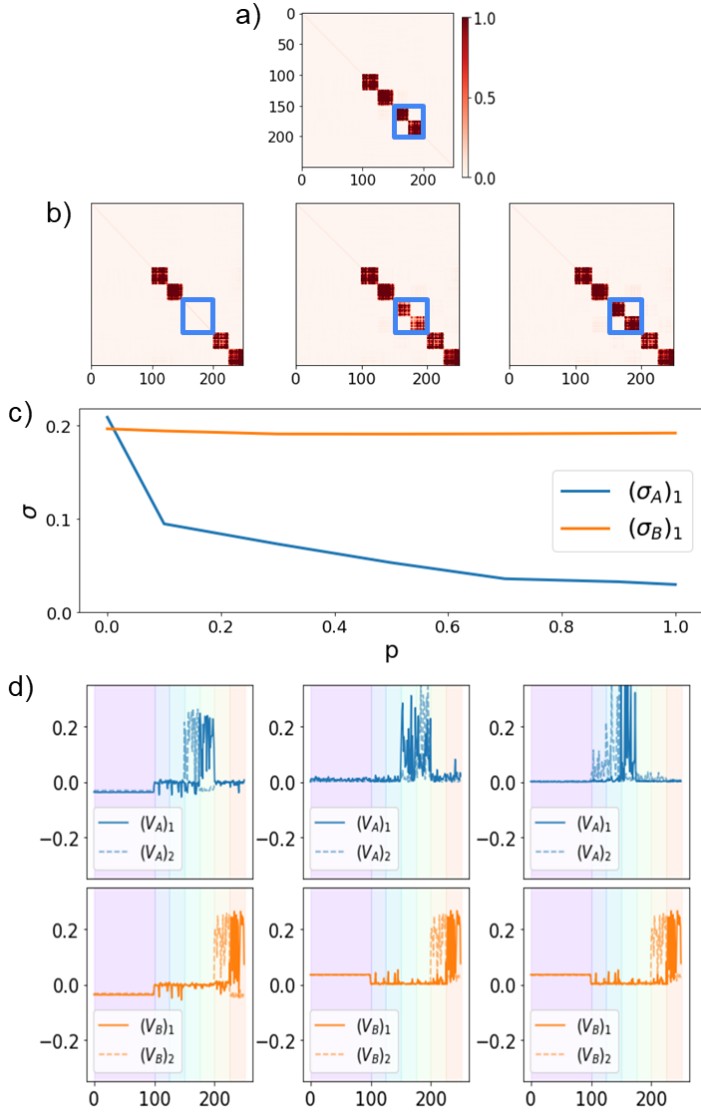

Figure 12: a) feature correlation matrix for dataset $A$. b) feature correlation matrix for dataset $B$ where p=0 (left), p=0.5 (middle) and p=1 (right). c) Significance of the first differential vector of $X^A$ (blue) and $X^B$ (orange). d) First two differential vectors of $X^A$ (top row) and $X^B$ (bottom row) for p=0 (left), p=0.5 (middle) and p=1 (right).

theoretical vs. numerical slope value is written over each panel. The numerical value matches the theoretical value almost perfectly.

Next, we repeat the same experiment to compare the numerical and theoretical increase for the numerator of Eq. (15). Since $\lambda_3$ is proportional to $s$, The theoretical increase of the numerator is $O(l^{0.5} + l^{(3\alpha/2-0.5)})$. Figure 15 shows the numerator vs. $l$ on a log-log scale. Similarly to Figure 14, for each panel, we write the numerical value of the slope, vs. the theoretical one. As expected, for low values of $\alpha$, the dominant term is $O(l^{0.5})$. For larger values of $\alpha$, the increase is slightly slower than expected from theory, implying that the bound on the numerator can be improved.

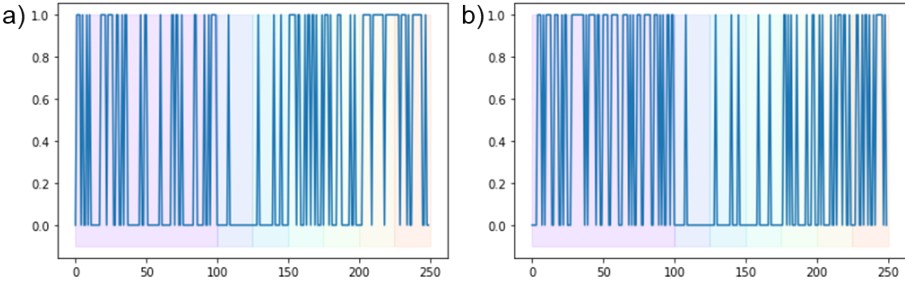

Figure 13: Iterative subset selection approach with NN (a) and logistic regression (b) classification accuracy criterion.

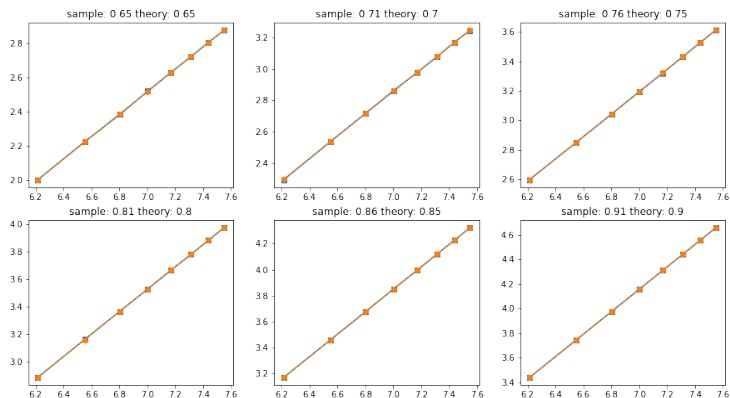

Figure 14: Caption: convergence of the denominator of Eq. (15)

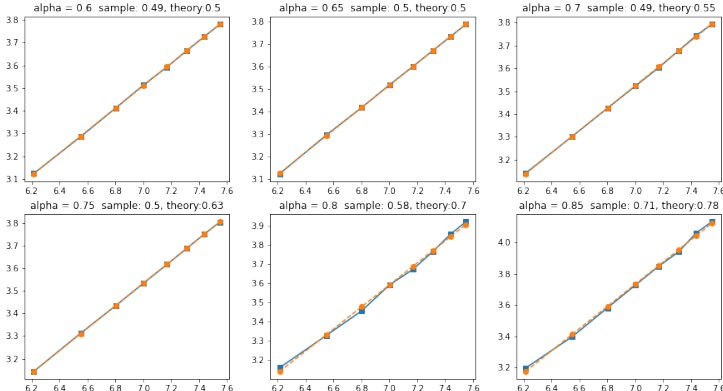

Figure 15: Caption: convergence of the numerator of Eq. (15)