# OpenReview forum: "DiSC: Differential Spectral Clustering of Features"
_NeurIPS.cc/2022/Conference — NeurIPS 2022 Accept_

### Official Review · Reviewer_T7Zb · 2022-07-11

**Rating:** 6
**Confidence:** 4
**Soundness:** 2 fair
**Presentation:** 2 fair
**Contribution:** 3 good

**Summary:**

The paper introduces a spectral approach for features which differentiate classes, in that these features are relevant to the structure in one class but not the other. The approach is based on producing a graph for each class which reflects the affinities between features in the associated class. The eigenvectors of the random walk (RW) Laplacian matrices of these graphs are generally seen as capturing the structure, and so in order to remove structure relevant to class B from the Laplacian from group A is projected into the null space of B's eigenvectors, and vice versa, and the spectral decomposition of these projected Laplacian matrices reveals structure relevant to the one class which is left after removing the relevant structure from the other. Post-processing of the singular vectors of the projected graph Laplacians provides groups of differentiating features (by clustering their rows) or meta-features (by projecting the original features from each class onto these singular vectors).

The paper provides a theoretical analysis in the context of a stochastic block model in which one group of features is shared by the two classes, while another is split in two by the one class but not the other.

The paper concludes with some interesting applications, showing potential for practical relevance of the method.

#### Update following author rebuttal and in light of other reviewers' comments
I am satisfied with the corrections which the authors have made which, I believe, remove my main concerns about the paper. The theoretical contributions are fairly modest and the proposed method not the most elegant, but I believe the novelty in how the authors have approached an interesting and potentially relevant problem is sufficient to justify acceptance. I think the work represents a good first investigation into this problem and I hope these authors, and potentially others, will improve on the formulation and solution of the problem in future.

**Questions:**

- Line 119: what is a "non-binary indicator vector"? and why is f one? Does this just mean the two distinct values are not 0 and 1?
- The formulation given in (4) and (5) seems more elegant than the one which is actually used. If this approach doesn't yield as good performance, then why discuss it in detail while omitting the detail for the approach which is actually employed?
- The constants in the lemmata are very important since, as mentioned above, all pairs of unit-length vectors are bounded in distance trivially. If the constants are large then the lemmata become irrelevant. Can you include them explicitly?
- What exactly does "with high probability" mean? I am sure it is not difficult to bring in this probability explicitly.
- I had to look at the appendix to learn what the quantities d_A and d_B in line 227 are. Do they actually appear in the paper somewhere?



I will also include in this section a few editorial comments/corrections
- LASSO is an acronym, and so is usually written capitalised, and not as Lasso.
- Notation of l-p norms should be consistent. Sometimes these are in math font, while other times not.
- line 64: "... identify groupS..."
- The term "independent components" carries a specific meaning in feature extraction, and I would suggest choosing a different phrase.
- Line 125: peviously "mincut" was "minimum-cut"
- The notation used for l-p norms is again not consistent in that sometimes "||.||" is given without the value p, whereas in at least one instance we find "||.||_2"
- The expression in (7) seems incorrect, since the argmax is taken over E, meaning that the vector (V_A)_i to which it is assigned should be a subspace, and not the vector v as desired.
- The two data sets are sometimes denoted X_A and X_B and sometimes X^A and X^B
- "algorithm 1" should be "Algorithm 1"
- the bracket in line 227 does not close.


**Limitations:**

I do not see any potential for foreseeable negative societal impact arising from the work.

**Strengths And Weaknesses:**

Strengths:
- The problem studied in the paper is a very interesting one, and the approach taken appears to be fairly novel, both at a high level (looking specifically for discriminating features) and at a low level (the specific methodology proposed).
- The presentation is fairly clear, and the paper well written for the most part.

Weaknesses:
- The major weakness of the paper is in its technical clarity and, potentially, correctness. In particular, unless I am missing something, the proof of Theorem 1 is not correct. For the sake of font, let v be the vector \tilde v_\gamma from the paper and e the vector \frac{1}{\sqrt{s}}e_\gamma from the paper. The authors seem to assert that an index, say i, for which |v(i) - e(i)| >= 1/2\sqrt{s} contributes at least 1/2\sqrt{s} to the l_2 distance between the vectors. However, I don't see how this is the case. All I can see is that the proof shows us ||v - e|| < C => ||v - e||^2 < C^2 => C^2 >= \sum_{i=1}^n (v(i) - e(i))^2 >= \sum_{misclassification indices} (v(i) - e(i))^2 >= (number of misclassifications) (1/2\sqrt{s})^2 = (number of misclassifications)/4s => (number of misclassifications) < 4sC^2. Indeed, my suspicion was raised by the fact that ANY two vectors of unit length are bounded in distance by 2, and so their argument seems to give us that a randomly sampled vector on the unit sphere, thresholded as they propose, would give a misclassification rate of at most 2\sqrt{s}. If I am mistaken in my interpretation of the proof of theorem 1, perhaps the authors can address this issue?

- I will point out other minor issues in the following section, via queries.

---

> ### Author Response · Authors · 2022-08-02
> **Proof of Theorem 1, Mathematical formulation**
>
> ## Proof of theorem 1
>
> We thank the reviewer for notifying us of the error found in the proof. Thankfully, after going over the different parts and lemmas, there is a simple fix. The main change is in Lemma 2, where by better handling the different terms we get rid of the constant factor, and remain with two factors that are proportional, respectively to $\frac{\sqrt{l}}{s}$ and $\sqrt{\frac{s}{l}}$.
>
> Thus, if we assume that the size s is proportional to $l^\alpha$, where $0.5<\alpha<1$, the difference between $v_\gamma$ and $\tilde v_\gamma$ decreases to 0 as $l \to \infty$. We changed Theorem 1 to match this observation.  The new Theorem and lemmas are provided in the updated paper. We are now working on extensive simulations to verify our theoretical findings - specifically, we want to explore weather the rate of decrease of $||\tilde v_\gamma -e_\gamma\|$ as a function of $l$ is indeed optimal or weather we can still improve the bound. We  will add the simulations to the paper to strengthen the Theorem.
>
> ## Clarification on $f$ in Eq. (3)
>
> We define $f$ in Eq. (3). The elements of $f$ consists of only two different values $\sqrt{\frac{|\beta|}{|\alpha|}}$ or $-\sqrt{\frac{|\alpha|}{|\beta|}}$, where the different values match the ratio of number of elements within each of the 2 groups. We will clarify this in the revised version.
>
> ## Performance of Eq. (4) and Eq.(5) compared to Eq. (7)
>
> The graph-cut perspective as detailed in (4) and (5) is given as motivation for our approach. The main idea is that while the graph-cut yields a densely connected component in a graph, the ‘differential graph-cut searches for densely connected components that appear only in one graph. It is similar in nature to the relation between the diffusion maps and differential vectors we compute. The leading diffusion map vectors are correlated with latent parameters that are associated with the underlying structure of the data [1] (division into clusters, trajectoreis, etc).
> In contrast, the differential vectors are correlated with latent parameters only of elements of the structure that appears in one of the graphs. We will clarify this point in the new version.
> As we explain on lines 155-158, we prefer the differential vectors over the graph cut (Eq. (5)) as the latter has some disadvantages, such as lack of stability (e.g., the need to add a regularizer for inverting the Laplacian) and the need to compute the inverse of the matrix. We hope this clarifies this point, if the reviewer feels that there are still missing details of our explanation of the differential vectors, we will be happy to elaborate.
>
> [1] Belkin, Mikhail, and Partha Niyogi. "Laplacian eigenmaps for dimensionality reduction and data representation." Neural computation 15.6 (2003): 1373-1396.
>
>
> ## Constants in lemmas
>
> We now fixed the proof so that the terms go to zero for large $l$. We can also change the constants such that the dependency on p,q is clear. That will still leave us with a constant C that is due to the concentration inequality of the norms $||W_A-E[W_A]||$ and $||W_B-E[W_B]||$. We cannot get rid of this constant as it is part of the standard concentration inequalities we apply, see for example Theorem 4.4.5 in [1]. However, this constant does not depend on any of the parameters of the problem. We ran some simulations with block sizes between 500 ~5000 to get a rough estimate, and the constant is around 1.5.
>
> [1] Vershynin, Roman. High-dimensional probability: An introduction with applications in data science. Vol. 47. Cambridge university press, 2018.
>
> ## Clarification regarding “with high probability” in lemmas.
>
> We added the explicit term for the probability, see Theorem 1 and Lemmas 1,3 in the new version. The probability is at least $1-\exp(-l)$. The randomness is due to the norms  of $||W_A-E[W_A]||$ and $||W_B-E[W_B]||$, see Theroem 4.4.5 in [1] (replace t with $\sqrt{n}$)
>
> [1] Vershynin, Roman. High-dimensional probability: An introduction with applications in data science. Vol. 47. Cambridge university press, 2018.
>
>
> ## Clarification on $d_A$ and $d_B$
>
> We define $d_B$ in line 144, where we define the projection matrix $Q_B$. We can make the definition more clear.
>
> We also thank the reviewer for the editorial comments. We will address these in the camera-ready paper.
>
> We hope we have addressed the reviewer's concerns and if so, we would appreciate if
> our score could be reevaluated. We are happy to clarify any additional concerns.

---

> > ### Author Response · Authors · 2022-08-04
> > **Simulations to verify theoretical findings**
> >
> > As mentioned in the previous comment, we generated simulations based on the double stochastic block model, which we analyzed in the paper. We now have results we intend to add to the new version. Our goal is to verify the bound we present in lemma 2 - $\|\|v_\gamma-\tilde v_\gamma \|\|$. The results illustrate the convergence of this term to zero at a rate that is exponential in the block size $l$, which matches our theory.  Recall that the size $s$ of the group $\gamma$ is equal to $l^\alpha$, where $0.5<\alpha<1$. The rate of convergence with respect to $l$ matches our theory exactly for moderate values of alpha (up to around 0.8), but the convergence rate seems faster than our bound for high levels of alpha - which implies it might be possible to tighten the bound.

---

> > > ### Comment · Reviewer_T7Zb · 2022-08-08
> > > **Nice**
> > >
> > > ... not much more to say

---

> > ### Comment · Reviewer_T7Zb · 2022-08-08
> > **Main concerns overcome**
> >
> > Thanks to the authors for their solid rebuttal to my comments.
> >
> > 1. Proof of Theorem 1:
> > That looks much better, thanks for the good work!
> > 2. Clarification of f:
> > Thanks, that's what I assumed
> > 3. Performance of (4) and (5):
> > I think it is worth mentioning the limitations, especially the instability and requirement of a regulariser. Any persuasive justification for the switch is valuable, since, as I mentioned, (4) and (5) seem more elegant in the absence of anything additional.
> > 4. Constants and w/h/p:
> > Fine, these were essentially addressed by fixing Theorem 1
> > 5. d_B:
> > Fine, and my apologies if I simply missed it.

---

> > > ### Author Response · Authors · 2022-08-08
> > > **Projection vs. dual objective**
> > >
> > > Thank you, we appreciate your feedback.
> > > We agree that a persuasive argument for the switch from (4) and (5) to (6) and (7) will strengthen the paper. We will add an explanation about the instability. We will add an experiment in the appendix to illustrate why the projection matrix better removes the shared effects.
> > > If you are satisfied with our response, we would appreciate it if you could reevaluate our score.

---

### Official Review · Reviewer_5Abs · 2022-07-12

**Rating:** 6
**Confidence:** 4
**Soundness:** 4 excellent
**Presentation:** 3 good
**Contribution:** 3 good

**Summary:**

This paper studies the selection of subsets of features that differentiate between conditions, which can be further used for classification or in-depth understanding of data or underlying mechanisms. It proposes a data-driven method DiSC to detect and group relevant features, which builds a graph with features as nodes and kernel functions as weights to quantify similarities between features. The principle is that the differences between class-specific graphs imply differences between conditions, and thus the task is to identify the independent components that appear in one graph but no the other. This paper further provides theoretical analysis for two stochastic block models and demonstrate the usefulness of DiSC in three toy datasets, MNIST, hyperspectral imagery, and fMRI datasets.


**Questions:**

As explained in Sec. "Weaknesses", we summarize the expected
improvements here:

1. More understanding of (7) in mathematics.

2. More toy examples or benchmark that clear our concerns.


**Limitations:**

We appreciate the authors' work that could potentially make a contribution. And we do not foresee any potential negative societal impact.


**Strengths And Weaknesses:**

Strengths:

1. This paper uses kernel functions to define the weights for class-specific graphs of feature similarities, which is a sound idea considering the flexibility of kernel methods.

2. A graph cut perspective is a brilliant idea, on which this paper is based to propose "differential vectors" and "significance level". They make a contribution by the property of non-symmetry and the increase of stability.

Weaknesses:

1. We expect more in-depth theoretical understanding of strengths of your extensions, differential vectors & significance level, compared to diffusion map, double objective, etc. Indeed you have pointed out the advantages of the extension. We agree with your comments roughly in logic, however, a rigorous analysis would be more convincing. From 3.1 to 3.2 you have shown a logic chain to propose (7). As the main section of your method, we expect more mathematical comments (now we only have a vague understanding of strengths of (7)).

2. Admittedly your toy examples have shown the usefulness of your proposed approach, but for these toy examples. What we probably expect more is a kind of benchmark that could explore the boundary of your method. These boundaries might be conditions that guarantee the sound performance of your approach. In toy examples, the differences between conditions are "sharp": one correlated, one ideally independent. Indeed you have an underlying assumption (Line 101-104), while we like to see the performance when applying to non-ideal case (i.e. when the approximated 0 is not so ideally small, how the performance is compromised.). For example, would our approach work for such a scenario: one strongly correlated, one weakly correlated; and how such a metric that describes the comparative difference of correlations affect the performance? (The concern is similar to non-separable case in classification.) Analysis in theory is challenging; at least we like to see the performance in simulation.

Strengths(?) or Weaknesses(?):
Sec. 3.3 is indeed a good work in theory; however, we do not know how to evaluate its contributions considering the main story of this paper. One may refer more to other reviewers' comments on this section.

---

> ### Author Response · Authors · 2022-08-02
> **Mathematical formulation and simulations on non-ideal settings**
>
> ## Clarification on Eq. (5) and Eq. (7)
> We agree with the reviewer that a few steps can be added to make the formulation of (7) flow more clearly from (5). Due to the instability and computational complexity of the matrix inversion of the graph Laplacian $L_A$, we instead replace this operator with an operator that projects $L_B$ onto the complementary subspace of the leading eigenvectors of $L_A$, leading to the operator $Q_A$. In practice however, rather than using the graph Laplacians $L_A$ and $L_B$, we propose to use the random-walk diffusion operators $P_A$ and $P_B$ instead. These operators, used as the basis of diffusion maps, have been shown to be connected to spectral clustering in [1],[2]. This finally leads to solving a singular value problem as in (7), where the formulation is not symmetric (as in (5)) and leading to two sets of differential vectors and significance values.
> [1] Nadler, Boaz, et al. "Diffusion maps, spectral clustering and eigenfunctions of Fokker-Planck operators." Advances in neural information processing systems 18 (2005).
> [2] Nadler, Boaz, et al. "Diffusion maps, spectral clustering and reaction coordinates of dynamical systems." Applied and Computational Harmonic Analysis 21.1 (2006): 113-127.
> ## DiSC in partially correlated conditions (non-ideal case)
> We empirically analyze the performance of detecting differential features in a non-ideal case, where, say, correlated components in one dataset are partially correlated in other dataset. In section 4.1, we compared two ideal settings - features that are highly correlated in one dataset and are completely independent in another. In section E.4, we added an extension to this experiment to test non ideal cases. To that end, we introduce a parameter $p$ that determines the correlation level. Specifically, we set the eigenvalues of the covariance to decrease exponentially such that the $i^{th}$ eigenvalue $\lambda_i$ is proportional to $exp(-i*p)$. Thus, if $p$ is close to 0 - the decrease is very small, which results in a covariance matrix close to identity and thus no correlation. If p is high, then the decrease is fast which results in high correlation. In the experiment, we kept the covariance of features 151-200 in $X^A$ fixed with $p = 1$. For these features of $X^B$, we changed the values of $p$ between 0 and 1 and computed the differential vectors and  significance level for each value of $p$. As expected the significance level of the differential vectors of $X^A$ decreases as p gets close to 1. For intermediate levels, the significance level is still high.  The results are shown in Figure 12 in the new version of the paper.
>
> We hope we have addressed the reviewer's concerns and if so, we would appreciate if
> our score could be reevaluated. We are happy to clarify any additional concerns.

---

> > ### Comment · Reviewer_5Abs · 2022-08-08
> > **Response to the authors**
> >
> > Thanks for the additional experiments and more detailed clarifications.
> >
> > Your description of additional experiments on the performance in non-ideal cases sounds good, and could resolve our concerns.
> >
> > The proposal of your method has been well described, and has been clear. After skimming the references (here added and ref. in paper), it seems your contribution of proposing a "new" method could have been largely weaken. It seems the techniques have been soundly known. We are now concerned with your novelty and technical contributions in this paper. Probably we missed something.
> > (Probably a bit more in-depth theoretical analyses of your algorithm/method itself, which might show its technical novelty. The theoretical achievements in Sec.3.3 look roughly independent from your theme of DiSC algorithm.)
> > Due to this major concern, we temporarily decreased the score to 5.

---

> > > ### Author Response · Authors · 2022-08-08
> > > **Novelty and relevance of the double stochastic block model**
> > >
> > > This is a bit of surprising feedback, as to the best of our knowledge, our approach is entirely new. We are confident that it doesn't appear in any of the papers we cited. If you can direct us to a specific paper with related ideas, we would be happy to verify this point.
> > >
> > > Our approach is part of a family of methods that are based on spectral representations. Specifically, as we pointed out in Sec. 3, it is related to 'diffusion maps' - a dimensionality reduction method that effectively captures latent processes in high dimensional data. In the works by Nadler et al. we cited, they derive theoretical foundations for diffusion maps and its relation to spectral clustering. However, these papers address the standard setting, where the goal is to compute a low dimensional representation of *observations*. **However, It does not address the setting of a dual dataset, where the goal is to extract differential groups of ***features*** associated with processes that are unique to only one of the datasets**.
> > > Thus, our approach differs from diffusion maps in its formulation and algorithmic solution, which is illustrated in several of our experiments.
> > >
> > > With regard to the theoretical contribution, it is our opinion that these directly relate to our approach, as given in Eq. (6) and (7). There are indeed two minor differences in the formulation - we don't normalize the weight matrix, and the projection is taken from both sides, rather than only one. These changes make the resulting matrix symmetric, which simplifies the analysis.
> > >
> > > This example motivates our approach from spectral graph theory. The question is when is it possible to detect a densely connected independent component that appears only in one graph/dataset. That is, in the other graph, this component is part of a larger component and is non-independent. This model is actually observed in several practical scenarios. For example, in the hyperspectral imaging datasets, patches of pixels that are part of a background in one dataset (say they are part of the sky/grass) become an independent component in the second graph as they highlight objects that appear only in the second dataset. We will clarify this point in the revised version.
> > >
> > > We will be happy to clarify these points further if needed.

---

> > > > ### Comment · Reviewer_5Abs · 2022-08-10
> > > > **Reply to comments on novelty**
> > > >
> > > > Thanks for the further explanations. We agree that our previous comments underestimate the contribution of your algorithm. We have raised the score.

---

### Official Review · Reviewer_SwNk · 2022-07-12

**Rating:** 7
**Confidence:** 3
**Soundness:** 3 good
**Presentation:** 3 good
**Contribution:** 2 fair

**Summary:**

This paper proposes a closed-form spectral solution to the problem of selecting feature selection that maximize the difference between two datasets.

This paper was the hardest to review in my batch. While the approach is a rather straightforward application of spectral community detection on graphs, the theoretical results are interesting, and the problem rather practical. I put up my review a very low confidence with a hope to discuss the paper more with other reviewers/AC.


**Questions:**

Can you clarify the selection of the baselines in the paper?


**Limitations:**

N\A

**Strengths And Weaknesses:**

[Strengths]

The problem is very basic. Given two data sets, what makes them different? This seems like a rather fundamental tool in the data analysis toolbox to have, and I appreciate a closed-form solution.

Theoretical results on the stochastic block model are solid and interesting. They are informative to practical application of the method and provide interesting insights on distinguishability of differentiating features.

[Weaknesses]

The experimental study seems rather forced. The problem is rather basic, and I would love to see comparison to some straightforward baselines from the fields of nonparametric statistics, or even naive iterative selection strategies based on some notion of divergence in the data. The design of a MNIST experiment is very unclear to me – why do authors average pixel values for differentiating features? I find it hard to evaluate hyperspectral and fMRI experimental results as well.

---

> ### Author Response · Authors · 2022-08-02
> **Experimental section and choice of baselines for group feature selection**
>
> ## DiSC vs. iterative selection strategies
>
> The main focus of the paper is the task of group feature selection - that is, uncovering groups of features whose joint expression can effectively differentiate between two conditions. For example, in the presence of a subset of features that are highly correlated, our goal is to identify (ideally) all of them. This is in contrast to methods such as best subset selection or LASSO that are ‘sparsity oriented’; in the existence of a highly correlated group, they typically yield only a small subset. Thus, our baseline comparison is to methods such as Elastic-net, where the regularizer is designed to make coefficients of correlated features equal, and are thus more appropriate for our setting. In addition, we compare to diffusion maps due its theoretical and algorithmic connection to the differential vectors, as explained in Sec 3.2.
>
> To illustrate this point, we applied the best subset selection approach to the simulated example in section 4.1, Identifying newly connected features. We applied the iterative feature selection approach to select the top 100 features that best differentiate $X^A$ and $X^B$. For our criterion to the best subset selection, we used the accuracy of a nearest neighbor and logistic regression classifiers, that take as input the selected features. (As implemented and suggested in the sklean implementation). The accuracy was very poor compared to our group feature selection method, and we refer to the results in the updated supplementary material section E.5. We can also include similar experiments on the MNIST experiment in the camera-ready paper.
> ##  Experiments
> Differential vectors identify the groups of features that together differentiate one condition/dataset from others. In the case of MNIST, this corresponds to clusters of pixels that together differentiate between digits. This can be, for example, clusters of pixels that together form a line or an area that uniquely characterizes a digit. The average over differentiating features can be thought of as a simple ‘meta-feature’ that captures the statistics of the group of features.
> In the experiment, our goal was to showcase our advantage over other methods in our ability to
> detect groups of pixels that effectively discriminate between digits. To that end, we compared a classifier that is trained on the average of pixel groups detected by our approach, vs. the average of pixel groups detected by other methods such as elastic net.
>
> The fMRI and hyperspectral are included as real-world datasets. The hyperspectral experiment is designed to showcase our advantage for identifying only  the differentiating areas in the image. These are, for example, the added object that appears in the data ‘october change’ within the grass.
> We compare DiSC to ‘diffusion map on changing data [1]’ which is a spectral approach  designed to capture differences between two conditions. Though this method captures the differences (grass and tarp) it also captures many more irrelevant areas in the image.
> For the fMRI experiments we have time-series recordings from 268 brain regions (parcels) for 515 subjects. These recordings can be split into two datasets corresponding to two separate tasks (0-back and 2-back) the subjects performed multiple times in the experiment. We treat the brain regions as features and aim to identify groups of brain features with correlated patterns. To evaluate these groups we compare them (via correlation) to 10 canonical brain networks of the brain regions. DiSC is able to identify for a group of brain regions corresponding to the frontal-parietal network that distinguishes the 2-back task from the 0-back task. This is a meaningful result since this network has been shown to be predictive of working memory which is required in the 2-back task. DIffusion maps on the other hand only highlights the visual networks in both tasks which is a trivial result. We will clarify this in the final version of the paper.
> ## Selection of baselines
> For MNIST, We compare our approach with diffusion maps, Elastic net and Elastic Net - logistic. We chose these approaches as they are designed to address cases of highly correlated features.
> For the hyperspectral imaging, simulated scRNA seq and MNIST, we compare to diffusion maps due to the connection explained in 3.2. Specifically, diffusion maps provide an effective spectral representation that highlights the main features in the dataset. In contrast, differential vectors are an effective spectral representation that highlight the main differential features in the dataset, while leaving out the features that are shared between the two datasets.
>
> We hope we have addressed the reviewer's concerns and if so, we would appreciate if our score could be reevaluated. We are happy to clarify any additional concerns.
>
> [1] Ronald R Coifman and Matthew J Hirn. Diffusion maps for changing data. Applied and computational harmonic analysis, 2014

---

> > ### Comment · Reviewer_SwNk · 2022-08-08
> > **Acknowledgement**
> >
> > Thanks for the discussion and additional clarifications on the experiments. I do think that the 2-sample classifier selection strategy on  MNIST would significantly strengthen the paper.
> >
> > Looking back, I was too harsh in the evaluation for some reason - sorry for that. I have raised my score to accept (7).

---

### Official Review · Reviewer_FFG2 · 2022-07-12

**Rating:** 4
**Confidence:** 3
**Soundness:** 2 fair
**Presentation:** 1 poor
**Contribution:** 2 fair

**Summary:**

This paper introduces a data-driven approach for detecting groups of features that differentiate between conditions. For each condition, we construct a graph whose nodes correspond to the features and whose weights are functions of the similarity between them for that condition. Then a spectral approach is applied to compute subsets of nodes whose connectivity differs significantly between the condition specific feature graphs.

**Questions:**

See weaknesses above.

**Ethics Review Area:**

["I don’t know"]

**Strengths And Weaknesses:**

Strengths:
1. The experimental results are good;
Weaknesses:
1. The proposed work makes me confused after I read the paper. It is not clear what the proposed method aims for, feature selection?
2. If the proposed work aims for feature selection, why uses two datasets A and B?
3. If the feature dimension p is very large, there will be a large number of nodes on the graph. How to handle the high computational cost?

---

> ### Author Response · Authors · 2022-08-02
> **Problem statement and computational complexity**
>
> ## Clarification about the problem statement
>
> Given 2 or more datasets with shared features or a dataset with multiple classes/conditions, we aim to identify groups of differential features; features that are highly correlated in one condition but not in others. That is, we develop a feature selection method for identifying groups of features whose correlation or connectivity pattern are different in one condition compared to another. Therefore we formulate the problem as there being two datasets A and B (for more than 2 see the appendix). A use case for such a method appears in the hyperspectral imaging dataset, where one would like to detect differences between two datasets captured at different times. Pixels that are part of such a difference will form an independent connected component only in the graph that corresponds to the datasets where the difference appears. In the second graph, these pixels will be part of a larger component.
> Another example is fMRI, where the goal is to detect patterns of connectivity between voxels that appear during one task, but not during a different task.
>
> ## Computational complexity
>
> Thank you for this question on computational costs. Our approach is graph-based. Naively computing a graph based on a kernel function is of order $O(p^2)$. For some cases, this can be prohibitively large. However, there are simple ways to reduce the complexity to be close to linear. For example, a common approach is to compute the kernel function only for the closest k-nearest neighbors, and not for all the $O(p^2)$ pairs of features. Computing k-nearest neighbors for all features can be done effectively with structures such as KD-trees, with an average complexity (for each feature) of $k log(p)$. Thus the total complexity of computing such a graph is $O(kplog(p))$. More efficient graph construction is possible with approximate nearest neighbors.  A second advantage of this approach is that the Laplacian matrix is sparse, with only $kp$ non zero elements. Thus, the computation of the eigenvectors can be done efficiently as well with complexity $O(kp)$. We will add a discussion on computation complexity to the revised version.
>
> We hope we have addressed the reviewer's concerns and if so, we would appreciate if
> our score could be reevaluated. We are happy to clarify any additional concerns.

---

### Author Response · Authors · 2022-08-07
**Feedback from reviewers**

We have responded to all reviewers. If there are additional questions that we should address, we would be happy to do so. Rev. SwNk  and Rev. T7Zb reduced our scores due to very specific points in the paper, which we fixed. We would appreciate if they can reevaluate their scores based on our response. In any case, any feedback from the reviewers will be appreciated.

---

### Meta-Review · Area_Chair_ArFm · 2022-08-23

**Recommendation:** Accept
**Confidence:** Certain

**Metareview:**

In the discussion, the area chair and the reviewers reached a consensus that the potential for practical relevance of the proposed method is very interesting and agreed to recommend the acceptance of the paper. The reviews provide valuable feedback to the authors to improve the final version of their paper that we are looking forward to see at the conference.

**Award:**

No

---

### Decision · Program_Chairs · 2022-09-14

Accept